# Aligning Diffusion Models by Optimizing Human Utility

Shufan Li[1]†        Konstantinos Kallidromitis[2]†        Akash Gokul[3]†*        Yusuke Kato[2]

Kazuki Kozuka[2]

[1]University of California, Los Angeles
[2]Panasonic AI Research
[3]Salesforce AI Research
†Equal contribution * Work done outside of Salesforce.
Correspondence to jacklishufan@cs.ucla.edu

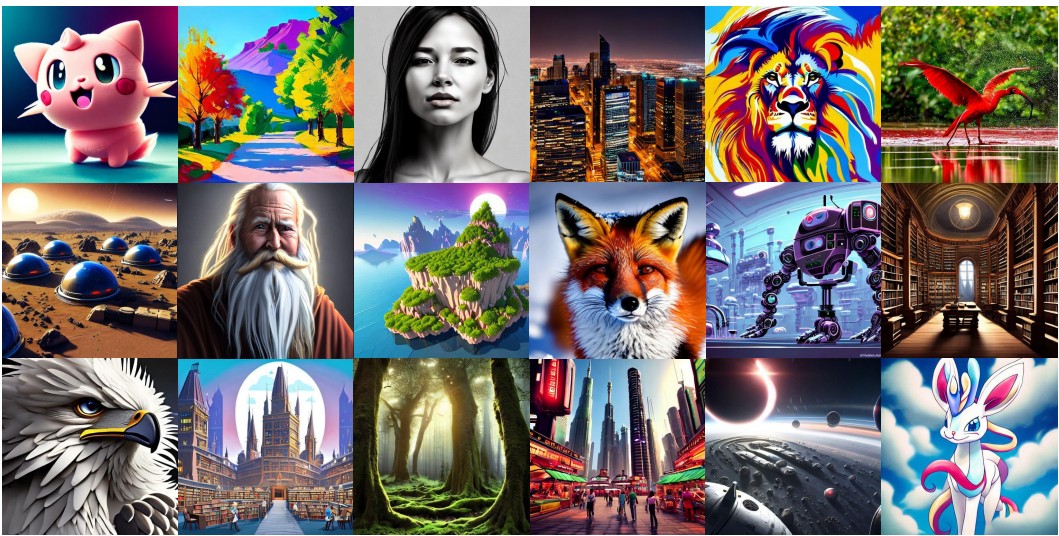

**Figure 1: Diffusion-KTO is a novel framework for aligning text-to-image diffusion models with human preferences using only per-sample binary feedback**. Diffusion-KTO bypasses the need to collect expensive pairwise preference data and avoids training a reward model. As seen above, Diffusion-KTO aligned text-to-image models generate images that better align with human preferences. We display results after fine-tuning Stable Diffusion v1-5 and sampling prompts from HPS v2 [50], Pick-a-Pic [27], and PartiPrompts [54] datasets.

## Abstract

We present Diffusion-KTO, a novel approach for aligning text-to-image diffusion models by formulating the alignment objective as the maximization of expected human utility. Unlike previous methods, Diffusion-KTO does not require collecting pairwise preference data nor training a complex reward model. Instead, our objective uses per-image binary feedback signals, *e.g.* likes or dislikes, to align the model with human preferences. After fine-tuning using Diffusion-KTO, text-to-image diffusion models exhibit improved performance compared to existing techniques, including supervised fine-tuning and Diffusion-DPO[48], both in terms of human judgment and automatic evaluation metrics such as PickScore [27] and ImageReward [52]. Overall, Diffusion-KTO unlocks the potential of leveraging readily available per-image binary preference signals and broadens the applicabil-

38th Conference on Neural Information Processing Systems (NeurIPS 2024).

ity of aligning text-to-image diffusion models with human preferences. Code is available at https://github.com/jacklishufan/diffusion-kto

# 1  Introduction

In the rapidly evolving field of generative models, aligning model outputs with human preferences remains a paramount challenge, especially for text-to-image (T2I) models. Large language models (LLMs) have made significant progress in generating text that caters to a wide range of human needs, primarily through a two-stage process: first, pretraining on noisy web-scale datasets, then fine-tuning on a smaller, preference-specific dataset. This fine-tuning process aims to align the generative model's outputs with human preferences, without significantly diminishing the capabilities gained from pretraining. Extending this fine-tuning approach to text-to-image models offers the prospect of tailoring image generation to user preferences, a goal that has remained relatively under-explored compared to its counterpart in the language domain.

Recent works have begun to explore aligning text-to-image models with human preferences. These methods either use a reward model and a reinforcement learning objective [52, 33, 14], or directly fine-tune the T2I model on preference data [48, 53]. However, these methods are restricted to learning from pairwise preference data, which consists of pairs of preferred and unpreferred images generated from the same prompt.

While paired preferences are commonly used in the field, it is not the only type of preference data available. Per-sample feedback is a promising alternative to pairwise preference data, as the former is abundantly available on the Internet. Per-sample feedback provides valuable preference signals for aligning models, as it captures information about the users' subjective distribution of desired and undesired generations. For instance, as seen in Fig. 2, given an image and its caption, a user can easily say if they like or dislike the image based on criteria such as attention to detail and fidelity to the prompt. While paired preference data provides more information about relative preferences, gathering such data is an expensive and time-consuming process in which annotators must rank images according to their preferences. In contrast, learning from per-sample feedback can utilize the vast amounts of per-sample preference data collected on the web and increases the applicability of aligning models with user preferences at scale. Inspired by these large-scale use cases, we explore how to directly fine-tune T2I models on per-image binary preference data.

To address this gap, we propose Diffusion-KTO, a novel alignment algorithm for T2I models that operates on binary per-sample feedback instead of pairwise preferences. Diffusion-KTO extends the utility maximization framework shown in KTO [18] to the setting of diffusion models. Specifically, KTO bypasses the need to maximize the likelihood from paired preferences and, instead, directly optimizes an LLM using a utility function that encapsulates the characteristics of human decision-making. While KTO is easy to apply to large language models, we cannot immediately apply it to diffusion models as it would require sampling across all possible trajectories in the denoising process. Although existing works approximate this likelihood by sampling once through the reverse diffusion process, back-propagating over all sampling steps is extremely computationally expensive. To overcome this, we present a utility maximization objective that applies to each individual sampling step, circumventing the need for sampling through the entire reverse diffusion process.

Our main contributions are as follows:

- We generalize the human utility maximization framework used to align LLMs to the setting of diffusion models (Section 4).

- Our method, Diffusion-KTO, facilitates alignment from per-image binary feedback. Thus, introducing the possibility of learning from human feedback at scale using the abundance of per-sample feedback that has been collected on the Internet.

- Through comprehensive evaluations, we demonstrate that generations from Diffusion-KTO aligned models are generally preferred over existing approaches, as judged by human evaluators and preference models (Section 5).

In summary, Diffusion-KTO offers a simple yet robust framework for aligning T2I models with human preferences that greatly expands the utility of generative models in real-world applications.

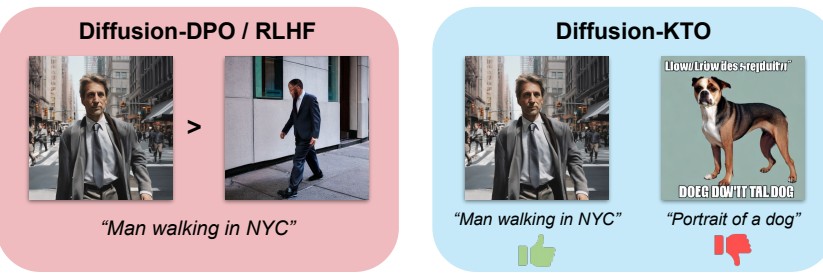

**Figure 2: Diffusion-KTO aligns text-to-image diffusion models using per-image binary feedback.** Existing alignment approaches (*Left*) are restricted to learning from pairwise preferences. However, Diffusion-KTO (*Right*) uses per-image preferences which are abundantly available on the Internet. As seen above, the quality of an image can be assessed independent of another generation for the same prompt. More importantly, such per-image preferences provide valuable signals for aligning T2I models, as demonstrated by our results.

## 2   Related Works

**Text-to-Image Generative Models.**  Text-to-image (T2I) models have demonstrated remarkable success in generating high quality images that maintain high fidelity to the input caption [38, 54, 37, 7, 11, 26, 55, 8]. In this work, we focus on diffusion models [43, 44, 24] due to their popularity and open-source availability. While these models are capable of synthesizing complex, high quality images after pretraining, they are generally not well-aligned with the preferences of human users. Thus, they can often generate images with noticeable issues, such as poorly rendered hands and faces. We seek to address these issues by introducing a fine-tuning objective that allows text-to-image diffusion models to learn directly from human preference data.

**Improving Language Models using Human Feedback.**  Following web-scale pretraining, large language models are further improved by fine-tuning on a curated set of data (supervised fine-tuning) and then using reinforcement learning to learn from human feedback. Reinforcement learning from human feedback (RLHF) [2, 12, 13], in particular, has been shown to be an effective means of aligning these models with user preferences [59, 5, 31, 30, 29, 45, 10, 3, 6, 22]. While this approach has been successful [1, 46], the difficulties in fine-tuning an LLM using RLHF [36, 58, 49, 17, 21, 42, 4] has led to the development of alternative fine-tuning objectives [35, 18, 56, 57]. Along these lines, KTO [18] introduces a fine-tuning objective that trains an LLM to maximize the utility of its output according to the Kahneman & Tversky model of human utility [47]. This utility maximization framework does not require pairwise preference data and only needs per-sample binary feedback. In this work, we explore aligning diffusion models given binary feedback data. As a first step in this direction, we generalize the utility maximization framework to the setting of diffusion models.

**Improving Diffusion Models using Human Feedback.**  Before the recent developments in aligning T2I models using pairwise preferences, supervised fine-tuning was the popular approach for improving these models. Existing supervised fine-tuning approaches curate a dataset using preference models [39, 32], pre-trained image models [41, 8, 16, 51, 50], and/or human experts [15], and fine-tune the model on this dataset. Similarly, many works have explored using reward models to fine-tune diffusion models via policy gradient techniques [19, 23, 9, 52, 14, 33, 28, 20] to improve aspects such as image-text fidelity. Similar to our work, ReFL [52], DRaFT [14], and AlignProp [33] align T2I diffusion models with human preferences. However, these methods require back-propagating the reward through the reverse diffusion sampling process, which is extremely expensive in memory. As a result, these works depend on techniques such as low-rank weight updates [25] and sampling from only a subset of steps in the reverse process, thus limiting their ability to fine-tune the model. In contrast, the Diffusion-KTO objective extends to each step in the denoising process, thereby avoiding such memory issues. More broadly, the main drawbacks of these reinforcement learning based approaches are: limited generalization, *e.g.* closed vocabulary [28, 20], reward hacking [14, 33, 9], and they rely on a potentially biased reward model. Since Diffusion-KTO trains directly on open-vocabulary preference data, we find that it can generalize to an open-vocabulary and avoids issues such as reward hacking. Recently, works such as Diffusion-DPO[48] and D3PO [53] present extensions of the DPO objective [35] to the setting of diffusion models. Diffusion-KTO shares

similarities with Diffusion-DPO and D3PO, as we build upon these works to introduce a reward model-free alignment objective. However, unlike these works, Diffusion-KTO does not rely on pairwise preference data and, instead, uses only per-image binary feedback.

## 3 Background

### 3.1 Diffusion Models

Denoising Diffusion Probabilistic Models (DDPM) [24] model the image generation process as a Markovian process. Given data $x_0$, the forward process $p(x_t|x_{t-1})$ gradually adds noise to an initial image $x_0$ according to a variance schedule, until it reaches $x_T \sim \mathcal{N}(0, \boldsymbol{I})$. A generative model can be trained to learn the reverse process $q_\theta(x_{t-1}|x_t)$ using the evidence lower bound (ELBO) objective:

$$\mathcal{L}_{\text{DDPM}} = \mathbb{E}_{x_0,t,\epsilon}[\lambda(t)\|\epsilon_t - \epsilon_\theta(x_t, t)\|^2] \tag{1}$$

where $\lambda(t)$ is a time-dependent weighting function and $\epsilon$ is the added noise.

### 3.2 Direct Preference Optimization

RLHF first fits a reward model $r(x, c)$, for a generated sample $x$ and input prompt $c$, to human preference data $\mathcal{D}$, and then maximizes the expected reward of a generative model $\pi_\theta$ while ensuring it does not significantly deviate from the initialization point $\pi_{\text{ref}}$. It uses the following objective with a divergence penalty controlled by a hyperparameter $\beta$.

$$\max_{\pi_\theta} \mathbb{E}_{c\sim\mathcal{D},x\sim\pi_\theta(x|c)}[r(x, c)] - \beta\mathbb{D}_{\text{KL}}[\pi_\theta(x|c)||\pi_{\text{ref}}(x|c)] \tag{2}$$

The authors of DPO [35] present an equivalent objective (Eq. (3)) using the implicit reward model $r(x, c) = \beta\log\frac{\pi_\theta(x|c)}{\pi_{\text{ref}}(x|c)} + \beta\log Z(c)$

$$\max_{\pi_\theta} \mathbb{E}_{x^w,x^l,c\sim\mathcal{D}}[\log\sigma(\beta\log\frac{\pi_\theta(x^w|c)}{\pi_{\text{ref}}(x^w|c)} - \beta\log\frac{\pi_\theta(x^l|c)}{\pi_{\text{ref}}(x^l|c)})] \tag{3}$$

where $Z(c)$ is the partition function, $(x^w, x^l)$ are pairs of winning and losing samples, and $c$ is the input conditioning. Through this formulation, the model $\pi_\theta$ can be directly trained in a supervised fashion without explicitly fitting a reward model.

### 3.3 Implicit Reward Model of a Diffusion Model

One of the challenges in adapting DPO to the context of diffusion models is that the likelihood $\pi_\theta(x|c)$ is hard to optimize because each sample $x$ is generated through a multi-step Markovian process. In particular, it requires computing the marginal distribution $\sum_{x_1...x_N}\pi_\theta(x_0, x_1...x_N|c)$ over all possible path of the diffusion process, where $\pi_\theta(x_0, x_1...x_N|c) = \pi_\theta(x_N)\prod_{i=1}^{N}\pi_\theta(x_{i-1}|x_i, c)$.

D3PO [53] adapted DPO by considering the diffusion process as a Markov Decision Process (MDP). In this setup, an agent takes an action $a$ at each state $s$ of the diffusion process. Instead of directly maximizing $r(x, c)$, one can maximize $Q(a, s)$ which assigns a value to each possible action $a$ at a given state $s$ in the diffusion process instead of the final outcome. This setup uses the local policy $\pi(a|s)$, which represents a single sampling step. In this setup, D3PO [53] showed that the optimal solution $Q^*(a, s)$ satisfies the relation $Q^*(a, s) = \beta\log\frac{\pi_\theta^*(a|s)}{\pi_{\text{ref}}(a|s)}$ for the optimal policy $\pi_\theta^*$. This leads to the approximate objective:

$$\max_{\pi_\theta} \mathbb{E}_{s\sim d^\pi, a\sim\pi_\theta(\cdot|s)}[Q(a, s)] - \beta\mathbb{D}_{\text{KL}}[\pi_\theta(a|s)||\pi_{\text{ref}}(a|s)] \tag{4}$$

where $d^\pi$ is the state visitation distribution under policy $\pi$. Concretely, the action is a sampling step, and we can write $Q(a, s)$ as $Q(x_{t-1}, x_t, c)$ and $\pi(a|s)$ as $\pi(x_{t-1}|x_t, c)$.

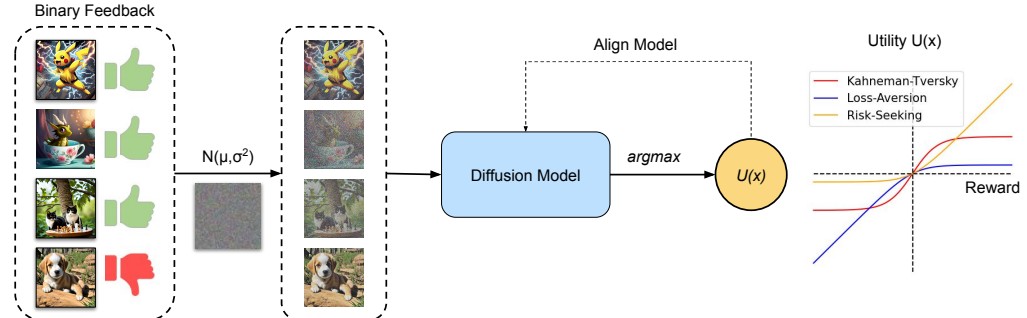

**Figure 3: We present Diffusion-KTO, which aligns text-to-image diffusion models by extending the utility maximization framework to the setting of diffusion models.** Since this framework aims to maximize the utility of each generation ($U(x)$) independently, it does not require paired preference data. Instead, Diffusion-KTO trains with per-image binary feedback signals, *e.g.* likes and dislikes. Our objective also extends to each step in the diffusion process, thereby avoiding the need to back-propagate a reward through the entire sampling process.

## 3.4 Kahneman-Tversky Optimization

In decision theory, the expected utility hypothesis assumes that a rational agent makes decisions based on the expected utility of all possible outcomes of an action, instead of using objective measurements such as the value of monetary returns. Formally, given an action $a$ and the set of outcomes $O(a)$, the expected utility is defined as $EU(a) = \sum_{o \in O(a)} p_A(o)U(o)$, where $p_A$ is a subjective belief of the probability distribution of the outcomes and the utility function $U(o)$ is a real-valued function.

Prospect theory [47] further augments this model by asserting that the utility function is not defined solely on the outcome (*e.g.* the absolute gain in dollars), but also with respect to some reference point (*e.g.* current wealth). In this formulation, the utility function is defined as $U(o, o_{\text{ref}})$ for a reference outcome $o_{\text{ref}}$. Based on this theory, KTO [18] proposed an alternative objective for aligning LLMs:

$$\max_{\pi_\theta} \mathbb{E}_{c,x \sim D}[\lambda(x)\sigma(\,w(x)(\beta \log \frac{\pi_\theta(x|c)}{\pi_{\text{ref}}(x|c)} - \mathbb{E}_{c' \sim D}\,[\beta\,\text{KL}(\pi_\theta(x'|c')\|\pi_{\text{ref}}(x'|c'))]))] \tag{5}$$

where $x$ is the output of the LLM, $c$ is the input prompt, $w(x) = 1$ if $x$ is desirable and $w(x) = -1$ otherwise, and $\lambda(x)$ is a weighting function of samples. The divergence penalty is computed as the expectation of the KL divergence between the model distribution $\pi_\theta(x'|c')$ and the reference distribution $\pi_{\text{ref}}(x'|c')$ over all input prompts $c'$ in the dataset. This formulation uses the sigmoid function $\sigma(x)$ as an approximation for the Kahneman-Tversky utility function which is concave in gain and convex in loss. Experiments showed that KTO aligned LLMs were able to outperform DPO aligned LLMs, and KTO is resilient towards noise in the preference data [18].

## 4 Method

### 4.1 Diffusion-KTO

Here, we propose Diffusion-KTO. Instead of optimizing the expected reward, we incorporate a non-linear utility function that calculates the utility of an action based on its value $Q(a, s)$ with respect to the reference point $Q_{\text{ref}}$.

$$\max_{\pi_\theta} \mathbb{E}_{s' \sim d^\pi, a' \sim \pi_\theta(\cdot|s)}[U(Q(a', s') - Q_{\text{ref}})] \tag{6}$$

where $U(v)$ is a monotonically increasing value function that maps the implicit reward to subjective utility. Practically, the local policy $\pi_\theta(a|s)$ is a sampling step, and can be written as $\pi_\theta(x_{t-1}|x_t)$. Applying the relation $Q^*(a, s)$ from Sec. 3.3, we get the objective

$$\max_{\pi_\theta} \mathbb{E}_{x_0 \sim \mathcal{D}, t \sim \text{Uniform}([0,T])}[U(\beta \log \frac{\pi_\theta(x_{t-1}|x_t)}{\pi_{\text{ref}}(x_{t-1}|x_t)} - Q_{\text{ref}})] \tag{7}$$

Following KTO [18], we can optimize the policy based on whether a given generation is considered "desirable" or "undesirable":

$$\max_{\pi_\theta} \mathbb{E}_{x_0 \sim \mathcal{D}, t \sim \text{Uniform}([0,T])}[U(w(x_0)(\beta \log \frac{\pi_\theta(x_{t-1}|x_t)}{\pi_{\text{ref}}(x_{t-1}|x_t)} - Q_{\text{ref}}))] \quad (8)$$

where $w(x_0) = \pm 1$ if image $x_0$ is desirable or undesirable. We set $Q_{\text{ref}} = \beta \mathbb{D}_{\text{KL}}[\pi_\theta(a|s)||\pi_{\text{ref}}(a|s)]$. Empirically, this is calculated by computing $\max(0, \frac{1}{m}\sum \log \frac{\pi_\theta(a'|s')}{\pi_{\text{ref}(a'|s')}})$ over a batch of unrelated pairs of $(s', a')$ following the KTO setup [18] in Eq. (5).

### 4.2 Utility Functions

While we incorporate the reference point aspect of the Kahneman-Tversky model, it is unclear if other assumptions about human behavior are applicable. It is also known that different people may exhibit different utility functions. Thus, we explore a wide range of utility functions. For presentation purposes, we center all utility functions $U(x)$ around 0 by using $U_{\text{centered}}(x) = U(x) - U(0)$, such that $U_{\text{centered}}(0) = 0$. This does not change the objective in Eq. (8) as the gradient and optimal policy are not affected. We experiment with the following utility functions:

- **Loss-Averse:** We characterize a loss-averse utility function as any utility function that is concave (see $U(x)$ plotted in blue in Figure 3). Using this utility function, the Diffusion-KTO objective can be considered as a variant of the Diffusion-DPO objective. While aligning according to this utility function follows a similar form to the Diffusion-DPO objective, our approach does not require paired preference data.

- **Risk-Seeking:** Conversely, we define a risk-seeking utility function as any convex utility function (see $U(x)$ plotted in yellow in Figure 3). A typical example of a risk-seeking utility function is the exponential function. However, its exploding behavior on $(0, +\infty)$ makes it hard to optimize. Instead, for this case, we consider $U(x) = -\log \sigma(-x)$.

- **Kahneman-Tversky model:** Kahneman-Tversky's prospect theory argues that humans tend to be risk-averse for gains but risk-seeking for losses relative to a reference point. This amounts to a function that is concave in $(0, +\infty)$ and convex in $(0, -\infty)$. Following the adaptation proposed in KTO, we employ the sigmoid function $U(x) = \sigma(x)$ (see $U(x)$ plotted in red in Figure 3). Empirically, we find this utility function to perform best.

Under the expected utility hypothesis, the expectation is taken over the subjective belief of the distribution of outcomes, not the objective distribution. In our setup, the dataset consists of unpaired samples $x$ that are either desirable ($w(x) = 1$) or undesirable ($w(x) = -1$). Because we do not have access to additional information, we assume the subjective belief of a sample $x$ is solely dependent on $w(x)$. During training, this translates to a biased sampling process where each sample is drawn uniformly from all desirable samples with probability $\gamma$ and uniformly from all undesirable samples with probability $1 - \gamma$.

## 5 Experiments

We comprehensively evaluate Diffusion-KTO through quantitative and qualitative analyses to demonstrate its effectiveness in aligning text-to-image diffusion models with a preference distribution. Further comparisons, such as the performance when using prompts from different datasets, the results of our ablations, and implementation and evaluation details can be found in the Appendix. Additionally, in the Appendix, we report the results of synthetic experiments which highlight that Diffusion-KTO can be used to cater T2I diffusion models to the preferences of a specific user. The code used for this work will be made publicly available and is available in the Supplementary material.

**Implementation Details.** We fine-tune Stable Diffusion v1-5 (SD v1-5) [39] (CreativeML Open RAIL-M license) with the Diffusion-KTO objective, using the Kahneman-Tversky utility function, on the Pick-a-Pic v2 dataset [27] (MIT license). The Pick-a-Pic dataset consists of paired preferences in the form of (preferred image, non-preferred image, input prompt). Since Diffusion-KTO does not require paired preference data, we partition the images in the training data. If an image is labelled

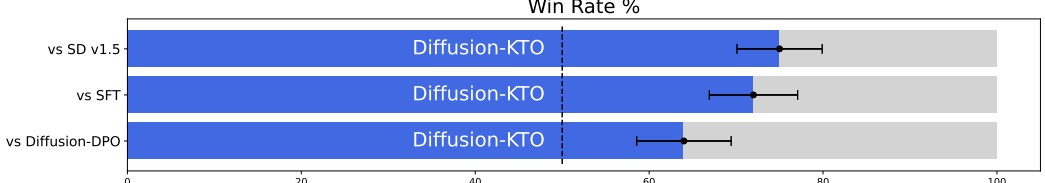

**Figure 4: User study win-rate (%) comparing Diffusion-KTO (SD v1-5) to SD v1-5, and SFT (SD v1-5) and Diffusion-DPO (SD v1-5)**. Results of our user study show that Diffusion-KTO significantly improves the alignment of the base SD v1-5 model. Moreover, our Diffusion-KTO aligned model also outperforms supervised finetuning (SFT) and the officially released Diffusion-DPO model, as judged by users, despite only training with simple per-image binary feedback. We also include the 95% confidence interval of the win-rate.

**Table 1: Automatic win-rate (%) for Diffusion-KTO (SD v1-5) in comparison to existing alignment approaches using prompts from the Pick-a-Pic v2 test set**. We use off-the-shelf models, *e.g.* preference models such as PickScore, to compare generations and determine a winner based on the method with the higher scoring generation. Diffusion-KTO drastically improves the alignment of the base SD v1-5 and demonstrates significant improvements in alignment when compared to existing approaches. Win rates above 50% are **bolded**.

| Method | Aesthetic | PickScore | ImageReward | CLIP | HPS v2 |
|---|---|---|---|---|---|
| vs. SD v1-5 | **86.0** | **85.2** | **87.2** | **62.0** | **62.0** |
| vs. SFT | **56.4** | **72.8** | **64.8** | **64.8** | **54.6** |
| vs. CSFT | **50.6** | **73.6** | **65.2** | **62.8** | **60.4** |
| vs. AlignProp | **86.8** | **96.6** | **84.4** | **96.2** | **90.2** |
| vs. D3PO | **68.0** | **73.6** | **71.6** | **56.8** | **55.6** |
| vs. Diffusion-DPO | **74.2** | **61.8** | **78.4** | **53.2** | **51.6** |

as preferred at least once, we consider it a desirable sample, otherwise we consider the sample undesirable. In total, we train with 237,530 desirable samples and 690,538 undesirable samples.

**Evaluation Details.** We evaluate the effectiveness of Diffusion-KTO by comparing generations from our Diffusion-KTO aligned model to generations from existing methods using automated preference metrics and user studies. For our results using automated preference metrics, we present win-rates (how often the metric prefers Diffusion-KTO's generations versus another method's generations) using the LAION aesthetics classifier [40] (MIT license), which is trained to predict the aesthetic rating a human would give to the provided image, CLIP [34] (MIT license), which measures image-text alignment, and PickScore [27] (MIT license), HPS v2 [50] (Apache-2.0 license), and ImageReward [52] (Apache-2.0 license) which are caption-aware models that are trained to predict a human preference score given an image and its caption. Additionally, we perform user studies to compare Diffusion-KTO with existing baselines. In our user study, we ask judges to assess which image they prefer (*Which image do you prefer given the prompt?*) given an image generated by our Diffusion-KTO model and an image generated by the other method for the same prompt.

We compare Diffusion-KTO to the following baselines: Stable Diffusion v1-5 (SD v1-5), supervised fine-tuning (SFT), conditional supervised fine-tuning (CSFT), AlignProp [33], D3PO [53] and Diffusion-DPO [48]. Our SFT baseline fine-tunes SD v1-5 on the subset of images that are labelled as preferred using the standard denoising objective. Our CSFT baseline, similar to the approach introduced into HPS v1 [51], appends a prefix to each prompt ("good image", "bad image") and fine-tunes SD v1-5 using the standard diffusion objective while training with preferred and non-preferred samples independently. To compare with D3PO (MIT license), we fine-tune SD v1-5 using their officially released codebase. For AlignProp (SD v1-5) (MIT license) and Diffusion-DPO (SD v1-5) (Apache-2.0 license), we compare with their officially released checkpoints.

## 5.1 Quantitative Results

Table 1 provides the win-rate, per automated metrics, for Diffusion-KTO aligned SD v1-5 and the related baselines. Diffusion-KTO markedly improves alignment of SD v1-5, with win-rates of up to 87.2%. Results from our user study (Figure 4) confirm that human evaluators consistently prefer the

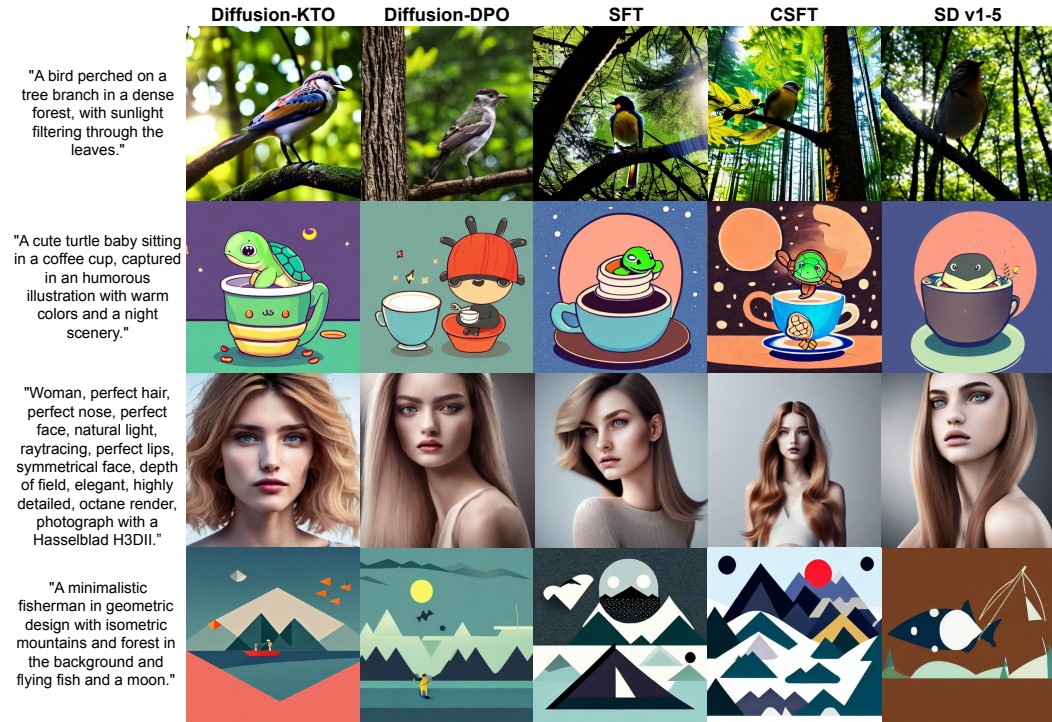

| Diffusion-KTO | Diffusion-DPO | SFT | CSFT | SD v1-5 |

**Figure 5: Side-by-side comparison of images generated by related methods using SD v1-5.** Diffusion-KTO demonstrates a significant improvement in terms of aesthetic appeal and fidelity to the caption (see Sec. 5.2).

generations of Diffusion-KTO to that of the base SD v1-5 (75% win-rate in favor of Diffusion-KTO). Further, Diffusion-KTO aligned models outperform related alignment approaches such as AlignProp, D3PO, and Diffusion-DPO. Diffusion-KTO significantly outperforms Diffusion-DPO on metrics such as LAION Aesthetics, PickScore, and HPS v2 while performing comparably in terms of other metrics. We also find that human judges prefer generations from our Diffusion-KTO model (72% win-rate versus SFT and 69% win-rate versus Diffusion-DPO) over that from SFT and Diffusion-DPO. This highlights the effectiveness of our utility maximization objective and shows that not only can Diffusion-KTO learn from per-image binary feedback, but it can also outperform models training with pairwise preference data.

## 5.2 Qualitative Results

In Fig. 5, we showcase a visual comparison of Diffusion-KTO with existing approaches for preference alignment. As seen in the first row, most models are misguided by the "sunlight" reference in the prompt and produce in a dark image. Diffusion-KTO demonstrates a focus on the bird, which is the central object in the caption and provides a better quality result over Diffusion-DPO, which doesn't include any visual indication for the "sunlight". In the second row of images, our Diffusion-KTO aligned model is able to successfully generate a *"turtle baby sitting in a coffee cup"*. Methods such as Diffusion-DPO, in this example, have an aesthetically pleasing result but ignore key components of the prompt (*e.g. "turtle","night"*). On the other hand, SFT and CSFT follow the prompt but provide less appealing images. For the third prompt, which is a detailed description of a woman, the output from our Diffusion-KTO model provides the best anatomical features, symmetry, and pose compared to the other approaches. Notably, for this third prompt, the generation of the Diffusion-KTO model also generated a background is more aesthetically pleasing. The final row uses a difficult prompt that requires a lot of different objects in a niche art style. While all models were able to depict the right style, *i.e.* geometric art, only the Diffusion-KTO generation includes key components such as the "moon," "flying fish," and "fisherman" objects. These examples demonstrate that Diffusion-KTO significantly increases the visual appeal of generated images while improving image-text alignment.

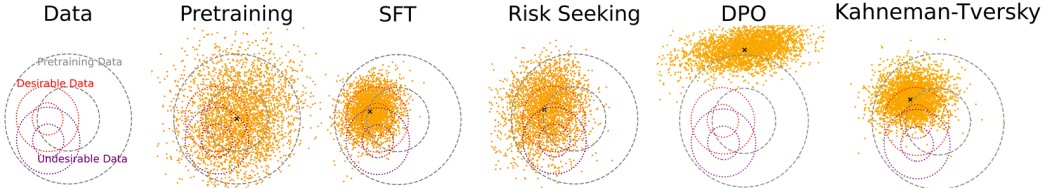

**Figure 6: Visualizing the effect of various utility functions.** We sample from MLP diffusion models trained using various alignment objectives. We find that using the Kahneman-Tversky utility function leads to the best performance in terms of aligning with the desirable distribution and avoiding the undesirable distribution.

# 6 Analysis

To further study the effect of different utility functions, we conduct miniature experiments to observe their impact. We assume the data is two-dimensional, and the pretraining data follows a Gaussian distribution centered at (0.5, 0.8) with variance 0.04. We sample desirable samples from a Gaussian distribution $P_d$ centered at (0.3, 0.8), sample undesirable samples from a Gaussian distribution $P_u$ centered at (0.3, 0.6), and the variance of both distributions is 0.01. We pretrain small MLP diffusion models, using the standard diffusion objective on the pretraining data, and then fine-tune using various utility functions. We sample 3500 data points from the trained model. Figure 6 shows that the risk-averse utility function (used by Diffusion-DPO) has a strong tendency to avoid loss to the point that it deviates from the distribution of desirable samples. The risk-seeking utility function behaves roughly the same as the SFT baseline and shows a strong preference for desirable samples at the cost of tolerating some undesirable samples. In comparison, our objective achieves a good balance.

# 7 Limitations

While Diffusion-KTO significantly improves the alignment of text-to-image diffusion models, it suffers from the shortcomings of T2I models and related alignment methods. Specifically, Diffusion-KTO is trained on preference data from the Pick-a-Pic dataset which contains prompts submitted by online users and images generated using off-the-shelf T2I models. As a result, the preference distribution in this data may be skewed toward inappropriate or otherwise unwanted imagery. Furthermore, in this work, we examined three main models of human utility, from which we have concluded the Kahneman-Tversky model to perform best based on empirical results. However, we believe that the choice of utility function, as well as the underlying assumptions behind such functions, remains an open question. Additionally, since Diffusion-KTO fine-tunes a pretrained T2I model, it inherits the weaknesses of this model, including generating images that reflect and propagate negative stereotypes. Despite these limitations, Diffusion-KTO presents a broader framework for improving and aligning diffusion models from per-image binary feedback.

# 8 Conclusion

In this paper, we introduced Diffusion-KTO, a novel approach to aligning text-to-image diffusion models with human preferences using a utility maximization framework. This framework avoids the need to collect pairwise preference data, as Diffusion-KTO only requires simple per-image binary feedback, such as likes and dislikes. We extend the utility maximization approach, recently introduced to align LLMs, to the setting of diffusion models and explore various utility functions. Diffusion-KTO aligned diffusion models lead to demonstrable improvements in image preference and image-text alignment when evaluated by human judges and automated metrics. While our work has empirically found the Kahneman-Tversky model of human utility to work best, we believe that the choice of utility functions remains an open question and promising direction for future work.

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

# A Experiment Settings

## A.1 Implementation Details

We train Stable Diffusion v1-5 (SD v1-5) on 4 NVIDIA A6000 GPUs with a batch size of 2 per GPU using the Adam optimizer. We use a base learning rate of 1e-7 with 1000 warm-up steps for a total of 10000 iterations. We set $\beta$ to 5000. We sample from the set of desirable samples according to $\gamma = 0.8$. To make Diffusion-KTO possible on paired preference datasets such as Pick-a-Pic [27], we create a new sampling strategy where we categorize every image that has been labelled as preferred at least once as desirable samples and the rest as undesirable samples.

## A.2 Evaluation Details

We employ human judges via Amazon Mechanical Turk (MTurk) for our user studies. Judges are given the prompt and a side-by-side image consisting of two generations from two different methods (*e.g.* Diffusion-KTO and Diffusion-DPO [48]) for the given prompt. We gather prompts by randomly sampling 100 prompts, 25 from each prompt style ("Animation", "Concept-art", "Painting", "Photo"), from the HPS v2 [50] benchmark. We collect a total of 300 human responses for our user study. In the interest of human safety, we opt to use prompts from HPS v2 instead of Pick-a-Pic [27], as we have found some prompts in the latter to be suggestive or otherwise inappropriate. The authors of HPS v2 incorporate additional filtering steps to remove inappropriate prompts. We also inform judges that they may be exposed to explicit content by checking this box in the MTurk interface and including the disclaimer: "WARNING: This HIT may contain adult content. Worker discretion is advised" in our project description. We gauge human preference by asking annotators which image they prefer given the prompt, *i.e.* "*Which image do you prefer given the prompt?*". The instructions given to our judges are provided below. Judges are asked to select "1" or "2", corresponding to which image they prefer (1 refers to the image on the left and 2 refers to the image on the right, these values are noted above each image when displayed). To ensure a fair comparison, they are not given any information about which methods are being compared and the order of methods (left or right) is randomized. MTurk workers were compensated in accordance with the minimum wage laws of the authors' country. We follow the guidelines and approval required for such studies by our institution.

> **Instructions**
> Both of these images were generated by AI models trained to create an image from a text prompt. Which image do you prefer given the associated text?
> Example criteria could include: detail, art quality, aesthetics, how well the text prompt is reflected, lack of distortions/irregularities (e.g. extra limbs, objects). In general, choose which image you think you would consider to be "better".

For our evaluation using automated metrics, we report win-rates (how often the metric prefers Diffusion-KTO's generations versus another method's generations). Given a prompt, we generate one image from the Diffusion-KTO aligned model and one image using another method. The winning method is determined by which method's image has a higher score per the automated metric. To account for the variance in sampling from diffusion models, we generate 5 images per method and report the win-rate using the median scoring images. We evaluate using all the prompts from the test set of Pick-a-Pic, the test set of HPS v2, and the prompts from PartiPrompts.

The human evaluation experiment received exemption for IRB.

# B Additional Quantitative Results

## B.1 Performance in terms of Average Score per Preference Models.

In Table 2, we report the average score (and 95% confidence interval of the mean) given by each metric used in our automated evaluations. For each method (using SD v1-5), we sample 5 generations per prompt, for a total of 2500 generations (*i.e.* N=2500). As seen below, Diffusion-KTO exhibits state-of-the-art performance according to numerous metrics while performing comparably to the state-of-the-art in the remaining metrics. These results demonstrate the effectiveness of Diffusion-KTO for aligning T2I diffusion models with human preferences. Additionally, we report the 95% confidence interval of win rate in Table 3

Table 2: **Average score according to existing models when evaluated on the Pick-a-Pic test set**. We report the mean score and the 95% confidence interval of the mean when evaluating prompts from the Pick-a-Pic test set. Methods with the highest mean score according to a given metric are **bolded**.

| Method | Aesthetic | PickScore | ImageReward | CLIP | HPS v2 |
|---|---|---|---|---|---|
| SD v1-5 | 5.281 ±.022 | 20.387 ±.054 | 0.333 ±.002 | 31.364 ±.144 | 0.102 ±.042 |
| SFT | 5.499 ±.020 | 20.664 ±.052 | 0.336 ±.002 | 31.377 ±.141 | 0.485 ±.038 |
| CSFT | **5.527 ±.020** | 20.713 ±.053 | 0.335 ±.002 | 31.448 ±.147 | 0.488 ±.040 |
| AlignProp | 5.106 ±.021 | 19.123 ±.054 | 0.278 ±.002 | 26.932 ±.144 | 0.195 ±.042 |
| D3PO | 5.326 ±.022 | 20.413 ±.055 | 0.333 ±.002 | 31.350 ±.147 | 0.143 ±.042 |
| Diffusion-DPO | 5.380 ±.022 | 20.785 ±.055 | 0.339 ±.002 | 31.673 ±.145 | 0.293 ±.042 |
| Diffusion-KTO | **5.527 ± .020** | **20.908 ±.054** | **0.342 ±.002** | **31.781 ±.143** | **0.623 ±.039** |

Table 3: **Confidence Interval of Win Rate (%) on Pick-a-Pic test set**. We report the the 95% confidence interval of the win rate over 2500 samples.

| Method | Aesthetic | PickScore | ImageReward | CLIP | HPS v2 |
|---|---|---|---|---|---|
| vs. SD v1-5 | **86.0±1.36** | **85.2±1.39** | **87.2±1.31** | **62.0±1.90** | **62.0±1.90** |
| vs. SFT | **56.4±1.94** | **72.8±1.74** | **64.8±1.87** | **64.8±1.87** | **54.6±1.95** |
| vs. CSFT | **50.6±1.96** | **73.6±1.73** | **65.2±1.87** | **62.8±1.89** | **60.4±1.92** |
| vs. AlignProp | **86.8±1.33** | **96.6±0.71** | **84.4±1.42** | **96.2±0.75** | **90.2±1.17** |
| vs. D3PO | **68.0±1.83** | **73.6±1.73** | **71.6±1.77** | **56.8±1.94** | **55.6±1.95** |
| vs. Diffusion-DPO | **74.2±1.72** | **61.8±1.90** | **78.4±1.61** | **53.2±1.96** | **51.6±1.96** |

## B.2 Performance on HPS v2 and PartiPrompts.

In addition to the results on the Pick-a-Pic test set reported in Table 1, we provide additional results on HPS v2 (Apache-2.0 license) and PartiPrompts (Apache-2.0 license) datasets in Table 4 and Table 5. Results show that Diffusion-KTO outperforms existing baselines on a diverse set of prompts.

Table 4: **Automatic win-rate (%) for Diffusion-KTO in comparison to existing alignment approaches using prompts from the HPS v2 test set**. The provided win-rates display how often automated metrics prefer Diffusion-KTO generations to that of other methods. Win rates above 50% are **bolded**.

| Method | Aesthetic | PickScore | ImageReward | CLIP | HPS v2 |
|---|---|---|---|---|---|
| vs. SD v1-5 | **76.2±1.67** | **77.7±1.63** | **74.3±1.71** | **53.5±1.96** | **53.6±1.95** |
| vs. SFT | **60.1±1.92** | **66.6±1.85** | **54.5±1.95** | **54.9±1.95** | **51.9±1.96** |
| vs. CSFT | **51.5±1.96** | **64.7±1.87** | **52.2±1.96** | **54.5±1.95** | **55.3±1.95** |
| vs. AlignProp | **86.2±1.35** | **98.0±0.55** | **81.0±1.54** | **93.2±0.99** | **89.7±1.19** |
| vs. D3PO | **76.0±1.67** | **76.9±1.65** | **75.6±1.68** | **54.4±1.95** | **53.9±1.95** |
| vs. Diffusion-DPO | **63.9±1.88** | **60.3±1.92** | **66.3±1.85** | 47.3±1.96 | 48.9±1.96 |

In addition, we provide a per-style score breakdown using the prompts and their associated styles (Animation, Concept-art, Painting, Photo) in the HPSv2 test set in Appendix B.2. Across these

**Table 5: Automatic win-rate (%) for Diffusion-KTO in comparison to existing alignment approaches using prompts from PartiPrompts**. The provided win-rates display how often automated metrics prefer Diffusion-KTO generations to that of other methods. Win rates above 50% are **bolded**.

| Method | Aesthetic | PickScore | ImageReward | CLIP | HPS v2 |
|---|---|---|---|---|---|
| vs. SD v1-5 | **74.2±1.72** | **67.1±1.84** | **66.9±1.84** | **53.8±1.95** | **52.5±1.96** |
| vs. SFT | **55.0±1.95** | **65.0±1.87** | **53.6±1.95** | **54.4±1.95** | **53.2±1.96** |
| vs. CSFT | **50.9±1.96** | **62.3±1.90** | **53.9±1.95** | **51.8±1.96** | **53.0±1.96** |
| vs. AlignProp | **76.6±1.66** | **95.5±0.81** | **79.0±1.60** | **91.2±1.11** | **86.8±1.33** |
| vs. D3PO | **75.1±1.70** | **67.0±1.84** | **68.9±1.81** | **51.5±1.96** | **53.0±1.96** |
| vs. Diffusion-DPO | **66.2±1.85** | **52.7±1.96** | **56.4±1.94** | 49.6±1.96 | 46.1±1.95 |

metrics, our model performs best for "painting" and "concept-art" styles. We attribute this to our training data. Since Pick-a-Pic prompts are written by users, it will reflect their biases, e.g., a bias towards artistic content. Such biases are also noted by the authors of HPSv2 who state "However, a significant portion of the prompts in the database is biased towards certain styles. For instance, around 15.0% of the prompts in DiffusionDB include the name 'Greg Rutkowski', 28.5% include 'artstation'."

We also observe that different metrics prefer different styles. For example, the "photos" style has the highest PickScore but the lowest ImageReward. With this in mind, we would like to underscore that our method, Diffusion-KTO, is agnostic to the preference distribution (as long as feedback is per-sample and binary), and training on different, less biased preference data could avoid such discrepancies.

| Style | Aesthetic | PickScore | ImageReward | CLIP | HPS |
|---|---|---|---|---|---|
| anime | 5.493 | 21.569 | 0.716 | 34.301 | 0.368 |
| concept-art | 5.795 | 21.011 | 0.804 | 33.141 | 0.359 |
| paintings | 5.979 | 21.065 | 0.802 | 33.662 | 0.360 |
| photo | 5.365 | 21.755 | 0.471 | 31.047 | 0.332 |

**Table 6: Per-style score breakdown for different metrics in the HPSv2 test set.**

### B.3   Using Stable Diffusion v2-1.

We perform additional experiments, this time using Stable Diffusion v2-1 (SD v2-1) [39] (CreativeML Open RAIL++-M license). We fine-tune SD v2-1 using Diffusion-KTO with the same hyperparameters and compute listed in Appendix A. In Table 7, we compare Diffusion-KTO (SD v2-1) with SD v2-1 and Diffusion-DPO (SD v2-1). To compare with Diffusion-DPO, we fine-tune SD v2-1 using the official codebase released by the authors. As seen in Table 7, Diffusion-KTO outperforms the SD v2-1 base model and Diffusion-DPO according to most metrics while performing comparably in others. This highlights the generality of Diffusion-KTO, as it is an architecture agnostic approach to improving the alignment of any text-to-image diffusion model.

## C   Synthetic Experiment: Aligning with a Specific User

Per-image binary feedback data is easy to collect and is abundantly available on the internet in the forms of likes and dislikes. This opens up the possibility of aligning T2I models to the preferences of a specific user. While users may avoid the tedious task of providing pairwise preference, Diffusion-KTO can be used to easily align a diffusion model based on the images that a user likes and dislikes. Here, we conduct synthetic experiments to demonstrate that Diffusion-KTO can be used to align models to custom preference heuristics, in an attempt to simulate the preference of a select user.

We experiment using two custom heuristics to mock the preferences of a user. These heuristics are: (1) preference for red images (*i.e.* red filter preference) and (2) preference for images with high aesthetics score. For these experiments, we fine-tune Stable Diffusion v1-5 using the details listed in A.1. For the red filter preference experiment, we use (image, caption) pairs from the Pick-a-Pic

**Table 7: Automatic win-rate (%) for Diffusion-KTO when using Stable Diffusion v2-1 (SD v2-1).** The provided win-rates display how often automated metrics prefer Diffusion-KTO generations to that of other methods. Results using Diffusion-DPO (SD v2-1) were produced by training SD v2-1 with the Diffusion-DPO objective, using the official codebase released by the authors. Win rates above 50% are **bolded**.

| Dataset | Diffusion-KTO | Aesthetic | PickScore | ImageReward | CLIP | HPS v2 |
|---|---|---|---|---|---|---|
| Pick-A-Pic | vs SD v2-1 | **71.4** | **70.0** | **69.2** | **53.4** | 49.6 |
| | vs Diffusion-DPO | **63.8** | **67.4** | **66.2** | 50.0 | 44.8 |
| HPS v2 | vs SD v2-1 | **72.2** | **77.9** | **71.0** | **51.6** | **50.1** |
| | vs Diffusion-DPO | **65.8** | **71.4** | **67.3** | 49.9 | 47.4 |
| PartiPrompts | vs SD v2-1 | **69.2** | **67.2** | **65.0** | **50.8** | 48.0 |
| | vs Diffusion-DPO | **66.0** | **61.2** | **63.1** | 49.2 | 44.7 |

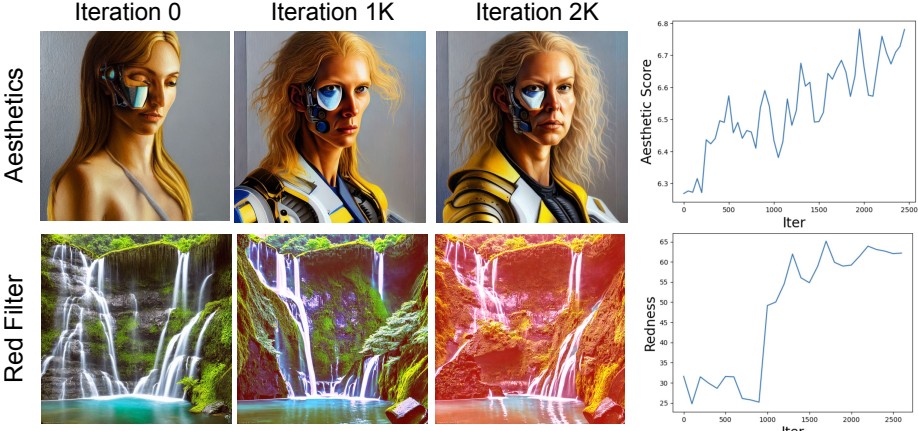

**Figure 7: Aligning text-to-image models with the preferences of a specific user.** Since per-image binary feedback is easy-to-collect, we perform synthetic experiments to demonstrate that Diffusion-KTO is an effective means of aligning models to the preferences of a specific user. We show qualitative results for customizing a text-to-image model with arbitrary user preference using Diffusion-KTO Stable Diffusion v1-5. The first row displays generations from a model that is trained to learn a preference for LAION aesthetics score $\geq 7$. As expected, these generations tend to introduce further detail (such as the woman's facial features) and add additional colors and textures. The second row displays images from a model that is trained to learn a preference for red images, and Diffusion-KTO learns to add this preference for red images while minimally changing the content of the image. We additionally plot the aesthetic score and redness score throughout the training. The redness score is calculated as the difference between the average intensity of the red channel and the average intensity in all channels.

v2 training set and enhance the red channel values to generate desired samples (original images are considered undesirable). For the aesthetics experiment, we train with images from the Pick-a-Pic v2 training set. We use images with an aesthetics score $\geq 7$ as desirable samples and categorize the remaining images as undesirable. Figure 7 provides visual results depicting how Diffusion-KTO can align with arbitrary user preferences. For the aesthetics preference experiment (Figure 7 row 1), we see that generations contain finer detail and additional colors and textures, in comparison to the baseline image, both of which are characteristics of a high scoring images per the LAION aesthetics classifier. Similarly, Diffusion-KTO also learns the preference for red images in the red filter preference experiment (Figure 7 row 2). While we experiment with simple heuristics, these results show the efficacy of Diffusion-KTO in learning arbitrary preferences using only per-sample binary feedback.

# D Ablations

We explored various design choices of Diffusion-KTO in this section. We report the mean PickScore on the HPS v2 dataset, which consists of 3500 prompts and is the largest amongst Pick-a-Pic, PartiPrompts, and HPS v2. We show results in Fig. 8

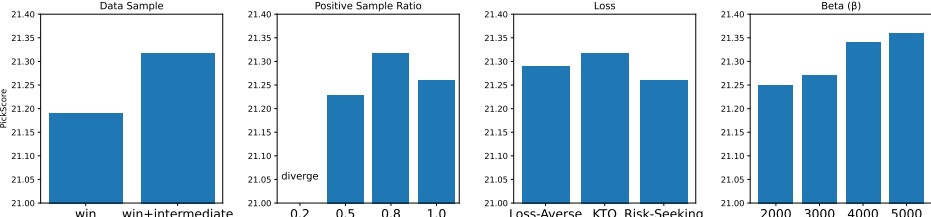

**Figure 8: Ablation Studies**. We experiment with different data sampling strategy and different utility functions. Results show the best combination is to a) use the winning (*i.e.* images that are always preferred) and intermediate (*i.e.* images that are sometimes preferred and sometimes non-preferred) samples, b) use a positive ratio of 0.8, c) use the KTO objective function, d) use a beta $\beta$ value of 5000.

## D.1 Data Partitioning

In converting the Pick-a-Pic dataset, which consists of pairwise preferences, to a dataset of per-image binary feedback, we consider two possible options. The first option is to categorize a sample as desired if it is always preferred across all pairwise comparisons (win), with all other samples considered undesirable. The second option additionally incorporate any samples as desirable samples if they are labelled as preferred in at least one pairwise comparison. Results show that the latter option is a better strategy.

## D.2 Data Sampling

During the training process, we sample some images from the set of all desired images and some images from the set of all undesired images. This ratio is set to a fixed value $\gamma$ to account for potentially imbalanced dataset. We find that sampling 80% of the images in a minibatch from the set of desired images achieves the optimal result.

## D.3 Choice of Beta

We experiment with different values between 2000 and 5000 for $\beta$, the parameter controlling the deviation from the policy. We show that there is a rise in performance with the increase in value but with diminishing returns, indicating an optimal score around 5000.

## D.4 Utility function

We explored the effect of various utility function described in the main paper. Particularly, we consider Loss-Averse $U(x) = \log \sigma(x)$, KTO $U(x) = \sigma(x)$ and Risk-Seeking $U(x) = -\log \sigma(-x)$. Results show that KTO is the optimal result.

We visualize the utility function and its first order derivative in Fig. 9. Intuitively, Loss-Aversion will reduce the update step during the training when reward is sufficiently high, Risk-Seeking will reduce the update step during the training when reward is low. KTO will reduce the update step if the reward is either sufficiently high or sufficiently low. This makes KTO more robust to noisy and sometimes contradictory preferences.

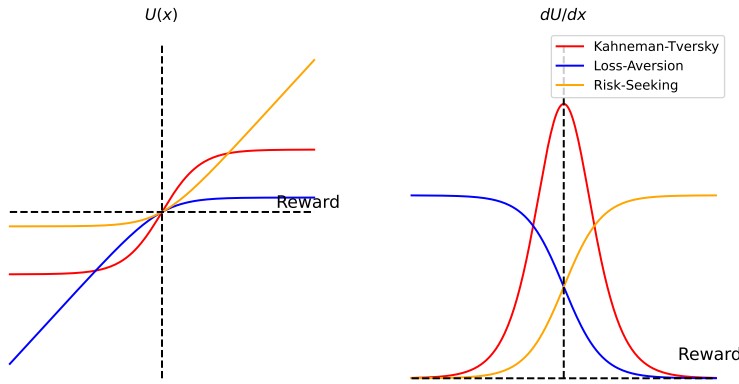

**Figure 9: Utility functions visualizations**. We visualize the utility function and its first order derivative. Intuitively, Loss-Aversion will reduce the update step during the training when reward is sufficiently high, Risk-Seeking will reduce the update step during the training when reward is low. KTO will reduce the update step reward is either sufficiently high or sufficiently low. The functions are centered by a constant offset so that $U(0) = 0$ for better visibility. The constant offset does not contribute to the gradient and, thus, has no effect to training.

## E  Additional Qualitative Results

We provide further visual comparisons between Diffusion-KTO aligned SD v1-5 and the off-the-shelf SD v1-5 (Fig. 10). We find that Diffusion-KTO aligned models improve various aspects including photorealism, richness of colors, and attention to fine details. We also provide visual examples of failure cases from Diffusion-KTO aligned SD v1-5 (Fig. 13).

## F  Safety

To understand the effect of aligning with Pick-a-Pic v2, which is known to contain some NSFW content, we run a CLIP-based NSFW safety checker on images generated using test prompts from Pick-a-Pic v2 and HPSv2. For Pick-a-Pic prompts, 5.4% of Diffusion-KTO generations are marked NSFW, and 4.4% of SDv1-5 generations are marked NSFW. For HPSv2 prompts, which are safer, 1.3% of Diffusion-KTO generations are marked NSFW, and 1.0% of SD v1-5 generations are marked NSFW. Overall, training on the Pick-a-Pic dataset leads to a marginal increase in NSFW content. We observe similar trends for Diffusion-DPO, which aligns with the same preference distribution (5.8% NSFW on Pick-a-Pick and 1.3% NSFW on HPSv2). We would like to emphasize that our method is agnostic to the choice of preference dataset, as long as the data can be converted into binary per-sample feedback. We used Pick-a-Pic because of its size and to fairly compare with related works. In general, we encourage fair and responsible use of our algorithm and

## G  Details of Qualitative Results

In this section, we discuss the sources of the prompts used in Fig. 5. To highlight the advantage of Diffusion-KTO in real-world scenarios, we refer to user prompts shared over the internet. In particular, we chose Playground AI (https://playground.com), where users share generated images alongside their prompts. These prompts reflect typical use cases in real-world scenarios and is generally similar to the "good" samples in the Pick-a-Pic dataset, which is also written by human labelers.

**Diffusion-KTO (SD v1-5)**     **SD v1-5**

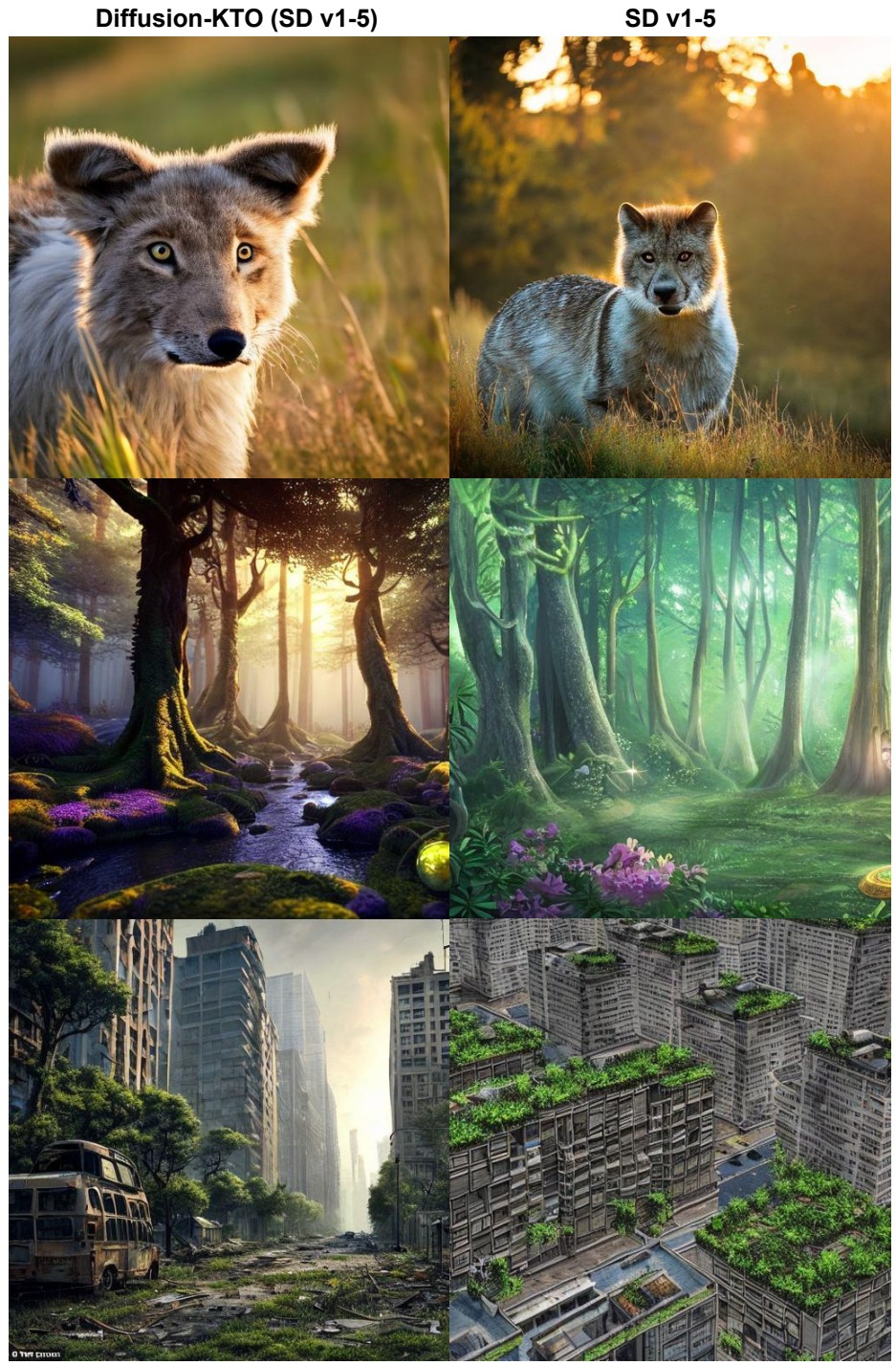

**Figure 10: Side-by-side comparison of Diffusion-KTO (SD v1-5) versus Stable Diffusion v1-5**. The images were created using the prompts: "A rare animal in its habitat, focusing on its fur texture and eye depth during golden hour.", "A magical forest with tall trees, sunlight, and mystical creatures, visualized in detail.", "A city after an apocalypse, showing nature taking over buildings and streets with a focus on rebirth."

**Diffusion-KTO (SD v1-5)**    **SD v1-5**

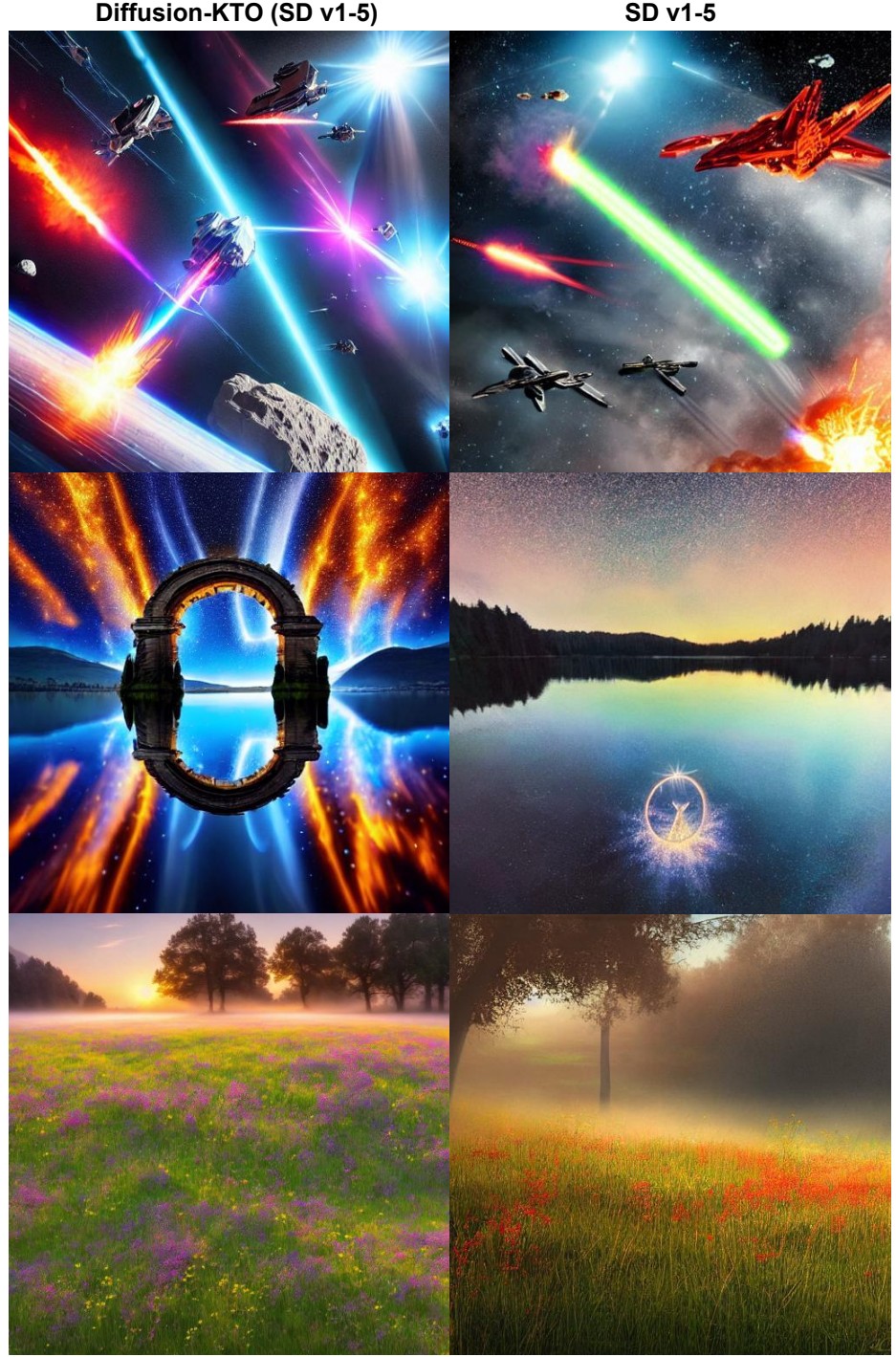

**Figure 11: Additional side-by-side comparison of Diffusion-KTO (SD v1-5) versus Stable Diffusion v1-5**.
The images were created using the prompts: "A dramatic space battle where two starships clash among asteroids,
with laser beams lighting up the dark void and explosions sending debris flying, intense and futuristic.", "A
timeworn portal in the middle of a serene lake, with glowing edges that ripple with energy, reflecting a starry sky
in its surface, captured in an ultra HD painting.", "A peaceful digital painting of a meadow at sunrise, where
wildflowers bloom and a gentle mist rises from the grass, soft focus."

**Diffusion-KTO (SD v1-5)**     **SD v1-5**

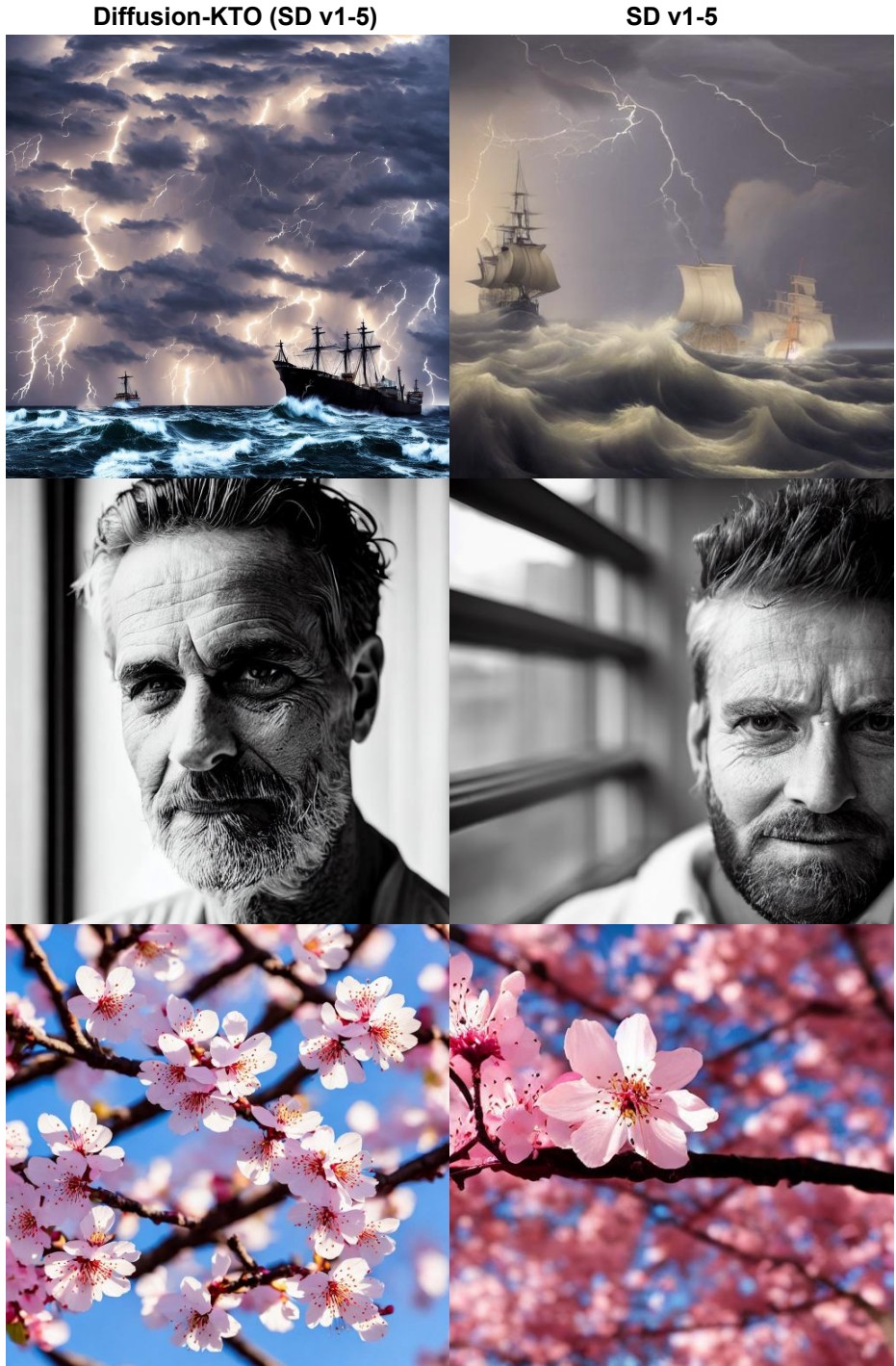

**Figure 12: Additional side-by-side comparison of Diffusion-KTO (SD v1-5) versus Stable Diffusion v1-5**. The images were created using the prompts: "A dramatic scene of two ships caught in a stormy sea, with lightning striking the waves and sailors struggling to steer, 8k resolution.", "A cinematic black and white portrait of a man with a weathered face and stubble, soft natural light through a window, shallow depth of field, shot on a Canon 5D Mark III.", "A hyperrealistic close-up of a cherry blossom branch in full bloom, with each petal delicately illuminated by the morning sun, 8k resolution."

"A playful dolphin leaping out of turquoise sea waves, with a backdrop of a stunningly colorful coral reef."

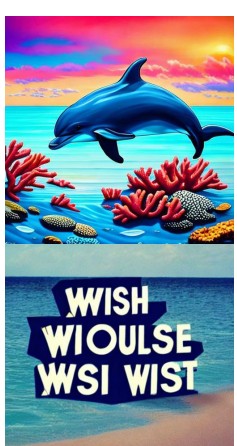

"An ancient dragon perched atop a mountain, overlooking a valley illuminated by the golden light of sunrise, with every scale visible in crisp detail."

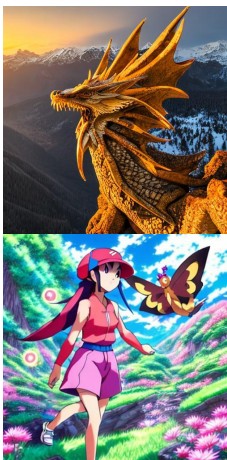

"Wish you were here"

"A Pokémon trainer discovering a valley with wild Pokémon with a Butterfree fluttering nearby and a group of Jigglypuff singing in the distance"

**Figure 13: Failures cases of Diffusion-KTO SD v1-5.** In the first instance (top-left) the dolphin is correctly shown "leaping out of the water", however, the coral reef is at the surface of the water not at the bottom of the sea. The second image (top-right) shows a dragon at the top of the mountain, however the dragon's body seems to merge with the stone. For the bottom left image, the caption is "Wish you were here" which is incorrectly written. The final image depicts a Pokémon trainer in a valley accurately, but it is missing a "group of Jigglypuff". We note that Diffusion-KTO aligned models inherit the limitations of existing text-to-image models.

