# OpenReview forum: "Aligning Diffusion Models by Optimizing Human Utility"
_NeurIPS.cc/2024/Conference — NeurIPS 2024 poster_

### Official Review · Reviewer_YybC · 2024-07-07

**Soundness:** 3
**Presentation:** 3
**Contribution:** 3
**Rating:** 7
**Confidence:** 4

**Summary:**

This paper introduces Diffusion-KTO, a novel approach for aligning text-to-image diffusion models with human preferences using per-image binary feedback (like likes/dislikes) rather than pairwise preference data. The key contributions include:

1. Extending the human utility maximization framework used to align language models to the domain of diffusion models.

2. Developing an objective that allows training on per-image binary feedback rather than pairwise preferences, enabling the use of abundant internet data like likes/dislikes.

3. Demonstrating through experiments that Diffusion-KTO outperforms existing alignment approaches like Diffusion-DPO, as judged by both automated metrics and human evaluators.

4. Showing that Diffusion-KTO can align models to specific user preferences using synthetic experiments.

The authors fine-tune Stable Diffusion models using their Diffusion-KTO objective on datasets like Pick-a-Pic. They evaluate the aligned models using automated metrics and human studies, comparing them against baselines like supervised fine-tuning and other alignment methods. The results indicate that Diffusion-KTO produces images that are preferred by humans and score higher on various automated metrics compared to existing approaches, while only requiring simpler per-image feedback data.

The paper also discusses limitations, broader impacts, and potential misuse of the technology. Overall, Diffusion-KTO presents a new framework for improving text-to-image models using readily available preference signals.

**Strengths:**

The paper demonstrates originality in several ways:
1. It extends the utility maximization framework from language models to diffusion models, representing a novel cross-domain application of ideas.
2. It introduces a new alignment objective that can work with per-image binary feedback rather than pairwise preferences, opening up new possibilities for using abundant internet data.
3, The approach allows for customization of text-to-image models to individual user preferences, which is an innovative direction in this field.

In terms of the quality of the approach and execution:
1. The authors provide a comprehensive evaluation, using both automated metrics and human studies to assess their method.
2. They compare against multiple baselines and state-of-the-art methods, demonstrating rigor in their experimental design.
3. The paper includes ablation studies.
4. The authors are transparent about limitations and potential negative impacts,

The paper is generally well-structured and clear:
1. The methodology is explained in detail, with the objective function presented.
2. Visual results are provided to illustrate the improvements, aiding in understanding.
3. The paper includes a detailed appendix with additional results and implementation details, supporting reproducibility.

The work is significant in how it improves text-to-image models in the setting when there is no preference data, but rather binary good/bad examples.  The ability to customize models to individual preferences could have broad implications for personalized AI systems.

**Weaknesses:**

The paper acknowledges that the method may learn biases from skewed preference data, but doesn't provide a detailed analysis of this issue. A more thorough examination of how different biases in the feedback data affect the aligned model would be valuable.

In some sense, the method itself is not novel as it has been just applied to the text-to-image setting where other preference-tuning methods have shown promise and similar methods from the NLP research can potentially be applied to this setting in the future, e.g. BCO or SimPO etc.

The experiments primarily focus on Stable Diffusion v1-5 and v2-1. Including a wider range of diffusion models, as they are generally available via the diffusers library would better demonstrate the generalizability of the approach.

While the paper demonstrates the potential for personalizing models to individual preferences, this is only shown through synthetic experiments.

**Questions:**

Did you use the trainers from the diffusers library to base your model on?

How does the effectiveness of Diffusion-KTO vary with different types of prompts? Is it more effective for certain categories of images or styles of prompts?

Have you considered extending the method to incorporate other types of feedback beyond binary signals? For instance, could it be adapted to use scalar ratings or textual feedback?

**Limitations:**

The paper includes a dedicated "Limitations" section (Section 7), which addresses several key points:
1. It acknowledges that Diffusion-KTO inherits limitations of the base text-to-image models.
2. It notes that the preference data used (Pick-a-Pic dataset) may contain biases or inappropriate content.
3. The authors recognize that the choice of utility function remains an open question.
4. They acknowledge that their model may propagate negative stereotypes present in the training data.

The authors discuss both potential positive and negative societal impacts:
Positive impacts:
1. Improved alignment of text-to-image models with human preferences.
2/ Potential for personalized image generation systems.

Negative impacts:
1. Risk of propagating biases present in preference data.
2. Potential for misuse in generating inappropriate or harmful imagery.
3. Concerns about the model inheriting weaknesses of the base text-to-image model, including generating images that reflect negative stereotypes.

The authors have made a good-faith effort to address limitations and societal impacts. They've covered key points such as data biases, potential misuse, and ethical considerations in experimentation.

---

> ### Author Rebuttal · Authors · 2024-08-06
>
> Thank you for the positive review! We are excited to hear that you found our method innovative and significant to improving text-to-image models. We are also happy to hear that you appreciated our comprehensive experiments and overall presentation. Please find our responses below.
>
>
> **A more thorough examination of how different biases in the feedback data affect the aligned model would be valuable.**
>
> To understand the effect of aligning with Pick-a-Pic v2, which is known to contain some NSFW content, we run a CLIP-based NSFW safety checker on images generated using test prompts from Pick-a-Pic v2 and HPSv2. For Pick-a-Pic prompts, 5.4% of Diffusion-KTO generations are marked NSFW, and 4.4% of SDv1-5 generations are marked NSFW. For HPSv2 prompts, which are safer, 1.3% of Diffusion-KTO generations are marked NSFW, and 1.0% of SD v1-5 generations are marked NSFW. Overall, training on the Pick-a-Pic dataset leads to a marginal increase in NSFW content. We observe similar trends for Diffusion-DPO, which aligns with the same preference distribution (5.8% NSFW on Pick-a-Pick & 1.3% NSFW on HPSv2). We would like to emphasize that our method is agnostic to the choice of preference dataset, as long as the data can be converted into binary per-sample feedback. We used Pick-a-Pic because of its size and to fairly compare with related works.
>
> **In some sense, the method itself is not novel... similar methods from the NLP research can potentially be applied to this setting in the future, e.g. BCO or SimPO etc.**
>
> Regarding BCO and SimPO, we agree that these works show promising results in the NLP domain. However we find it infeasible to implement and evaluate these methods for text-to-image generation in such a short rebuttal period. We will investigate the effectiveness of these approaches in future works. Please see the “Novel Contributions of Diffusion-KTO” of our main rebuttal for further details regarding novelty.
>
> **Including a wider range of diffusion models, as they would better demonstrate the generalizability of the approach.**
>
> We provide results using two different models (SD v1-5 and SD v2-1) in our main paper and Appendix. These results highlight the generality of Diffusion-KTO for various text-to-image diffusion models. We agree that it would be interesting to see how our method works for other state-of-the-art models. However, since recent models use more complicated architectures (e.g., Multi-Modal Diffusion Transformer of SD v3.0) and significantly more parameters, it is not feasible to complete these experiments within a week. We leave these endeavors for future work.
>
>
> **While the paper demonstrates the potential for personalizing models to individual preferences, this is only shown through synthetic experiments.**
>
> While binary preferences are easier to collect than paired preferences, curating high-quality personalized preference data is still expensive. We explored the possibility of scraping likes and dislikes from websites such as Artstation, but this does not comply with their terms of service. Additionally, it is hard to evaluate whether the model can effectively align with personalized preferences due to the diverse nature of human interests. In contrast, synthetic experiments are more controllable with measurable quantitative metrics such as aesthetic scores (Appendix C). For these reasons, we left the task of curating a high-quality personalized feedback dataset for future works.
>
> **Did you use the trainers from the diffusers library?**
>
> We refer the reviewer to our sample code in supplementary material for implementation details. We do not use the trainers.
>
> **How does the effectiveness of Diffusion-KTO vary with different types of prompts?**
>
> Below, we provide a per-style score breakdown using the prompts and their associated styles (Animation, Concept-art, Painting, Photo) in the HPSv2 test set. Across these metrics, our model performs best for "painting" and "concept-art" styles. We attribute this to our training data. Since Pick-a-Pic prompts are written by users, it will reflect their biases, e.g., a bias towards artistic content. Such biases are also noted by the authors of HPSv2 who state "However, a significant portion of the prompts in the database is biased towards certain styles. For instance, around 15.0% of the prompts in DiffusionDB include the name “Greg Rutkowski”, 28.5% include “artstation”." We also observe that different metrics prefer different styles. For example, the "photos" style has the highest PickScore but the lowest ImageReward. With this in mind, we would like to underscore that our method, Diffusion-KTO, is agnostic to the preference distribution (as long as feedback is per-sample and binary) and, training on different, less biased preference data could avoid such discrepancies.
>
> |    Style    | Aesthetic |  PickScore  | ImageReward |  CLIP  |  HPS  |
> |:-----------:|:---------:|:------:|:------------:|:------:|:-----:|
> |    anime    |   5.493   | 21.569 |     0.716    | 34.301 | 0.368 |
> | concept-art |   5.795   | 21.011 |     0.804    | 33.141 | 0.359 |
> |  paintings  |   5.979   | 21.065 |     0.802    | 33.662 | 0.360 |
> |    photo    |   5.365   | 21.755 |     0.471    | 31.047 | 0.332 |
>
> **Have you considered extending the method to incorporate other types of feedback beyond binary signals?**
>
> While we have considered exploring a variety of human feedback, this work, Diffusion-KTO, focuses specifically on binary feedback. For continuous feedback, we show in Appendix C an example where Diffusion-KTO can effectively align with the preference distribution after the continuous feedback signal is thresholded into a binary one. Textual feedback is non-trivial to implement in such a short rebuttal period, and we leave this for future work.

---

### Official Review · Reviewer_s9JR · 2024-07-09

**Soundness:** 3
**Presentation:** 3
**Contribution:** 4
**Rating:** 6
**Confidence:** 4

**Summary:**

This paper (DKTO) combines D3PO and KTO. D3PO let's us apply DPO to Diffusion models using pairwise preferences, and KTO is a way to align generative models (specifically autoregressive LLMs) using pointwise preference. For example, this gives us a way to tune text-conditioned, image generative models from thumbs-up, thumbs down or star ratings type of bandit feedback, without training a secondary reward model.

DKTO works by optimizing the expected utility of the advantage of the new policy versus a reference policy. And empirically, its results on finetuned Stable diffusion are pretty compelling. However, the mathematical derivation of DKTO is a bit messy and not rigorous.

**Strengths:**

The results in the paper are compelling and the contributions in the paper seem to be novel. AFAICT relevant baselines were considered for experiments and DKTO seems to win handily against them.

**Weaknesses:**

Section 4.1 which derives the actual diffusion KTO method is really short and seems wrong on first glance. To go from eq (6) to Eq(7) the relation Q* = β (log π_θ - log π_ref) is substituted but that relation is lifted from the D3PO paper that derived it under the assumption that the policy optimization objective is E[Q*] - β KL[π_θ || π_ref] . But the diffusion KTO objective is different so it's not clear why prop 1 from the DKTO paper still applies.

Secondly, the substitution Q_ref = β KL is not very well motivated. The notation Q_ref suggests that it should not have had any dependence on π_θ at all !! Why is the KL divergence a reasonable proxy for Q_ref ?

Overall, I think the paper in its current form is weak from a theory/conceptual point of view, and the method needs to be motivated a lot more cleanly. Please let me know if I have made a mistake, I'll gladly update my scores.

**Questions:**

See weaknesses.

**Limitations:**

Yes, authors have addressed the limitations adequately.

---

> ### Author Rebuttal · Authors · 2024-08-06
>
> Thank you for your review! We are glad that you appreciated the contributions and comprehensive experiments presented in this paper.
>
> Regarding the concerns on specific formulations, we have provided a detailed review of our formulation in the main rebuttal that should address these concerns. The relevant parts are summarized as follows:
>
> **It's not clear why prop 1 from the DKTO paper still applies.**
>
> We assume you are referring to prop1 from the D3PO (not DKTO) paper. It is sufficient to establish this relation under the policy optimization objective. The “substitution” is not wrong because KTO defines its implicit reward function $r_\theta(X_0,C)=\beta\log(\pi_{\theta}(X_0|C)/\pi_{ref}(X_0|C))$ in an axiomatic approach (Definition 3.4 of KTO paper).
>
> KTO justifies such definition through a “classic prospect theory experiment”, and the fact that $r_\theta$ is in the same equivalence class as the human-preference-induced reward function in (eq 2) at optimal policy $\pi^*_\theta$ under RLHF objective (eq 2). In prospect theory, $r_\theta$ "amounts to the dollar amount assigned to each outcome." However, the KTO formulation does not imply that the KTO objective is strictly equivalent to the RLHF objective in a similar way as DPO. There is no "substitution" in KTO formulation.
>
> By "applying the relation $Q^∗(a, s)$" in sec 4.1, we meant that we adopt the implied quality function definition $Q_\theta(X_{t-1},X_{t},C)=\beta\log(\pi_{\theta}(X_{t-1},X_{t},C)/\pi_{ref}(X_{t-1},X_{t},C)$ based on the relation $Q^*(X_{t-1},X_{t},C)= \beta\log(\pi^*_\theta(X_{t-1}|X_{t},C)/\pi_{ref}(X_{t-1}|X_{t},C))$, as well as the connection between $Q^*$ and the  RLHF objective established by D3PO.  We provide further details in the main rebuttal. If you have any additional questions, feel free to let us know and we are happy to answer them in discussion period.
>
> **Secondly, the substitution $Q_{ref}=\beta D_{KL}$ is not very well motivated.**
>
> The reference point of KTO is defined as $z_0=E_D[r_\theta(X_0,C)]$, which is the expected reward over some distribution $D$ of $(X_0,C)$. This definition stems from the assumption that "rather than having just one dispreferred generation serve as the reference point z0", "humans judge the quality of [generation] in relation to all possible input-output pairs."
>
> Similarly, we define $Q_{ref}$ as $E_{D'}[Q_\theta(X_{t-1},X_{t},C)]$ over some distribution $D'$ of $X_{t-1},X_{t},C,t$. KTO set $D$ to be a uniform distribution over the input dataset, and the reference point simplifies to the expected KL divergence. Following an identical derivation, we can establish that  $Q_{ref}$ in Diffusion-KTO formulation simplifies to the expected KL divergence.
>
> We provide further details in the main rebuttal. We will add these details in future versions and apologize for any confusion caused by this omission.
>
> Note: In the above discussion, we replace the term $l(y)$, the normalization factor in the definition of $r_\theta(X_0,C)$, with a constant $\beta$ following Eq(6) of KTO paper for simplicity. We also replace the notation $x,y$ to $C,X_0$ for consistency with notations of diffusion models.

---

### Official Review · Reviewer_7SUT · 2024-07-10

**Soundness:** 3
**Presentation:** 3
**Contribution:** 2
**Rating:** 5
**Confidence:** 4

**Summary:**

This paper presents Diffusion-KTO, a novel preference learning algorithm for diffusion models. The proposed preference learning algorithm is based on Kahneman & Tversky Optimization (KTO). Diffusion-KTO enables aligning a diffusion model using only binary pointwise feedbacks, improving data efficiency and robustness. The experiments show promising results.

**Strengths:**

* Aligning a diffusion model from human feedback is an important problem that receives huge attention from the community.
* The experimental results are promising and interesting.
* Overall presentation of the paper is clear, with some parts requiring improvements (see Weakness)

**Weaknesses:**

* The proposed algorithm is almost a simple application of KTO to diffusion models. The originality of the contribution is, therefore, not very strong. Of course, the combination of diffusion models and KTO is indeed novel; somebody else might have done it soon if this paper hadn't.  I acknowledge the novelty, but the paper would have been stronger with more original ideas.
* The paper has to be more self-contained. The paper borrows a lot of components from existing works, such as KTO and D3PO, and the borrowed elements should be explained well. Section 3.3 is particularly dissatisfying. There is no rigorous definition of Q, no definition of pi^*, and no explanation of why Equation 4 is an approximate objective. In a similar spirit, Section 4.1 could be more informative, such as providing an explanation for why Q_ref has to be set as proposed.
* The major concern regarding the experiment is that the number of human subjects and human responses are not disclosed (please correct me if I am wrong). This information is required to judge the uncertainty of the winning rate presented in Figure 4. It would be great if the uncertainty of the winning rate could also be provided.

**Questions:**

See weakness. I would adjust my rating based on the response regarding the weaknesses of the paper.

**Limitations:**

The paper addresses its limitations.

---

> ### Author Rebuttal · Authors · 2024-08-06
>
> Thank you for your review! We are happy to hear that you find our problem area important and that you appreciate the presentation of our work and our experimental results. Please see our responses below.
>
>
> **The proposed algorithm is almost a simple application of KTO to diffusion models.**
>
> Please see the “Novel Contributions of Diffusion-KTO” of our main rebuttal for our response.
>
>
> **The paper has to be more self-contained.**
>
> In regards to the specific concerns raised (i.e. the substitution of $Q$ and definition of $\pi^*$, and $Q_{ref}$), we provide a detailed review of our Diffusion-KTO formulation with clarifications for these definitions and choices in the main rebuttal (Clarification of Formulation). We hope these discussions can resolve your concerns. If you have any additional questions, please let us know and we would be happy to answer them in the discussion period.
>
>
> **The number of human subjects and human responses are not disclosed.**
>
> We are sorry for this oversight. We collected 300 human responses. The 95% confidence interval is 65.6%-74.8% for the win-rate against DPO and 73.6%-82.9% for the win-rate against SDv1-5. Both results are significant, with p-value < 0.001.

---

> > ### Comment · Reviewer_7SUT · 2024-08-11
> > **Thanks for reply.**
> >
> > Thanks for the response. Your answer addresses my questions well, and it is impressive to see the statistical significance of the result.

---

### Official Review · Reviewer_MiLc · 2024-07-12

**Soundness:** 3
**Presentation:** 3
**Contribution:** 3
**Rating:** 7
**Confidence:** 4

**Summary:**

This paper proposes Diffusion-KTO, which extends the KTO theory to develop a preference optimization algorithm for

**Strengths:**

* The derivations are clean, direct, and seem reasonably principled to me.

* The experimental results are good and show a clear improvement over preexisting work. However, I would include more quantitative examples in the main paper.

* Aligning diffusion model methods is a reasonably impactful area, and doing so with novel techniques is nontrivial and thus requires works like this one.

**Weaknesses:**

* While I enjoyed the paper, the novelty is somewhat limited. In particular, the paper is a relatively direct combination of the Diffusion DPO type methods and novel DPO style losses.

**Questions:**

* Can you include an example without the use of a preference pair dataset? Pick-a-pic is paired, but the main benefit of KTO should be the ability to extend beyond paired data (which might be useful for diffusion models since generation is expensive).

**Limitations:**

Yes.

---

> ### Author Rebuttal · Authors · 2024-08-06
>
> Thank you for your positive review! We are excited to hear that you appreciate the importance of our problem statement, the strength of our experimental results, and the presentation of our method. Please find our responses below.
>
> **The novelty is somewhat limited**
>
>
> Please see the “Novel Contributions of Diffusion-KTO” of our main rebuttal for our response.
>
> **Can you include an example without the use of a preference pair dataset**.
>
> We provide results when using two protocols for sampling at the population level of our data. We tried a) for each prompt in the dataset, incorporate both winner and loser separately as training data and b) for each prompt in the dataset, keep either the winner or loser as training data. Empirically, we find no significant differences between the performance of these two protocols. This effect is also seen by the authors of the original KTO paper. Specifically, we find the second protocol (b) performs slightly better on Aesthetics (+0.02), PickScore (+0.04), HPS (+0.002), CLIP (+0.13), and ImageReward (+0.05). These differences, apart from ImageReward, are within the error bound of estimates. The second setup (b) could be marginally better than the first setup (a) as it may remove some relative noise within paired preference data. In summary, we did not train with paired preferences and, we find that incorporating both the winner and loser separately or incorporating either the winner or loser into our training data generally leads to no significant difference in performance.
>
> Additionally, we would also like to point out that we tried two toy experiments using binary preferences in Appendix C. In these experiments, we created two synthetic datasets with binary labels by a) using a red filter and setting red-filtered images as desirable and non-filtered ones as undesirable, and b) thresholding the LAION aesthetic score and labeling samples with high scores as desirable and samples with low scores as undesirable. Results show that Diffusion-KTO can effectively align to binary preferences at the user level.

---

### Author Rebuttal · Authors · 2024-08-06

We thank the reviewers for their constructive feedback. We are glad to hear that the reviewers recognize our strong experimental results (Reviewers MiLc, 7SUT, s9JR, YybC), the importance of learning from binary preference data (Reviewers MiLc, 7SUT, YybC), and the novelty of extending the utility maximization framework to the setting of diffusion models (Reviewers s9JR & YybC).

In this main rebuttal, we address the common concern of novelty and clarify some common confusions on theory formulation. For other specific questions, we refer the reviewers to individual rebuttals.

**Novel Contributions of Diffusion-KTO**

We would like to emphasize the novel contributions of our work. Our work is the first to extend the utility maximization framework to the setting of diffusion models (as noted by Reviewers s9JR & YybC). Unlike the original KTO paper, we explore the effects of various utility functions and provide an analysis of their performance (see Section 6 and Appendix D.4). As we are extending this framework to a novel domain (text-to-image diffusion models), we perform such experiments to avoid naively assuming that the Kahneman & Tversky model would work best. Last but not least, we would like to highlight the potential impact of our method. Diffusion-KTO demonstrates that text-to-image diffusion models can be aligned with human preferences using binary preference data and can even outperform methods that require paired data. This is a new capability that, to our knowledge, was not previously available as the best alignment methods required paired preference data. As a result, this opens a world of possibilities as binary feedback is abundant on the internet and can be collected at scale.


**Clarification of Formulation**

We apologize for these oversights. A diffusion process from an image $X_0$ to random noise $X_T$ induces a conditional distribution $P(X_0|C)=\prod_{t=1}^{T}P(X_{t-1}|X_t)P(X_T|C)$ where C is the prompt. $P(X_T|C)$ can be written as $P(X_T)$ which is an i.i.d. Gaussian distribution. We can induce a global policy $\pi(X_0|C)$ representing the whole sample process, as well as a local policy $\pi(X_{t-1}|X_{t},C)$ representing each sample step. A reward function $r(X_0,C)$ is a real-valued function that encodes human preference. Every possible $r(X_0,C)$ induces an optimal global policy $\pi^*(X_0|C)$ that maximizes this reward. A "quality" function $Q(X_{t-1},X_{t},C)$ is a real-valued function that similarly induces an optimal local policy $\pi^*(X_{t-1}|X_{t},C)$.

Conceptionally, $Q$ can be an arbitrary real-valued function. $Q^*$ is a special choice of $Q$ such that the induced local policy maximizes the expected global reward $r(X_0,C)$.  Hence the (Eqn. 4) is an approximation of RL objective (Eqn. 2).  D3PO shows the relation $Q^*(X_{t-1},X_{t},C)= \beta\log(\pi^*_\theta(X_{t-1}|X_{t},C)/\pi_{ref}(X_{t-1}|X_{t},C))$. Following MDP convention, we can consider ($X_{t},C$) as a state $s$ and $X_{t-1}$ as an action $a$, and use $(s,a)$ in the notation instead. These formulations are elaborated in Sec 4.1 of the D3PO paper, we will incorporate more details in the future version of our paper.

In Diffusion-KTO, $Q$ and $Q_{ref}$ in (Eqn. 4) follow the definitions proposed in the original KTO paper. KTO adopted an axiomatic approach and defined the implicit reward function as $r_\theta(X_0,c)=\beta\log(\pi_{\theta}(X_0|C)/\pi_{ref}(X_0|C))$. The justification of this definition stems from classic prospect theory experiments, as well as the observation that $r_\theta$ is in the same equivalence class as the human-preference-induced reward function in (Eqn. 2) at optimal policy $\pi^*_\theta$ under RLHF objective (Eqn. 2).  This formulation can be found in Sec 3.2 of KTO paper. Unfortunately, this formulation is intractable as it involves the global policy $\pi(X_0|C)$, and cannot be directly applied.

Similarly, we can define implicit quality function $Q_\theta(X_{t-1},X_{t},C)=\beta\log(\pi_{\theta}(X_{t-1},X_{t},C)/\pi_{ref}(X_{t-1},X_{t},C)$, since results of D3PO have established the relation of $Q^*$ and the optimal policy.

The reference point of KTO is defined as $z_0=E_D[r_\theta(X_0,C)]$, which is the expected reward over some distribution $D$ of $(X_0,C)$. This definition stems from the assumption that "rather than having just one dispreferred generation serve as the reference point z0", "humans judge the quality of [generation] in relation to all possible input-output pairs."

Similarly, we define $Q_{ref}$ as $E_{D'}[Q_\theta(X_{t-1},X_{t},C)]$ over some distribution $D'$ of $X_{t-1},X_{t},C,t$. KTO set $D$ to be a uniform distribution over the input dataset, and the reference point simplifies to the expected KL divergence. Following an identical derivation, we can establish that  $Q_{ref}$ simplifies to the expected KL divergence. We will add these details in future versions and apologize for any confusion caused by this omission.

Note: In the above discussion, we replace the term $l(y)$, the normalization factor in the definition of $r_\theta(X_0,C)$, with a constant $\beta$ following Eq(6) of KTO paper for simplicity. We also replace the notation $x,y$ to $C,x_0$ for consistency with notations of diffusion models.

---

### Comment · Area_Chair_L2mV · 2024-08-07
**Could the demonstrated empirical gains overcome the concerns on novelty?**

Dear all,

From my assessment of the reviews and the rebuttal, there appears to be a prevailing concern regarding the novelty of the submission:

- The proposed Diffusion-KTO method is largely a straightforward extension of KTO, originally applied to fine-tuning language models, to fine-tuning diffusion models.
- It could also be viewed as a variant of Diffusion-DPO, where the DPO component is substituted with KTO.

Given these novelty concerns, the empirical performance improvements presented will be crucial in informing my decision. I do have reservations about the significance of these performance gains and would appreciate further insights from both reviewers and authors on this matter:

The preference data utilized is derived from the Pick-a-Pick training set. While Table 1 in the main text underscores the benefits of Diffusion-KTO, showing clearly higher winning rates based on Pick-a-Pick test prompts, the advantages are much less clear when examining Tables 3 and 4 in the appendix, which employ HPS v2 and Parti prompts, respectively. Are the differences in these tables still statistically significant? This ambiguity makes it challenging to ascertain whether Diffusion-KTO truly outperforms baseline methods, such as SFT that also does not require pairwise preferences.

Thanks,

AC

---

> ### Author Response · Authors · 2024-08-08
> **Response to statistical significance of HPSv2 and Parti prompts.**
>
> Thank you for recognizing our strong results on Pick-a-Pic. Regarding the results on HPS and PartiPrompts, these datasets have significantly more prompts than Pick-a-Pic, which leads to a smaller margin of error. For example, on HPS, all automatic win-rates above 51.4% have a p-value < 0.00024, and on PartiPrompts all automatic win rates above 51.4% have a p-value < 0.00729, both of which are below the threshold for statistical significance (p=0.05).
>
> We would also like to point out that our human evaluation is conducted on a subset of HPS prompts. The 95% confidence interval is 65.6%-74.8% for the human-evaluated win-rate against DPO and 73.6%-82.9% for the human-evaluated win-rate against SDv1-5. Both results are significant, with p-value < 0.001. These experiments reaffirm a key observation made by the authors of KTO when using KTO for LLMs: "margin between KTO and DPO is even bigger in human evaluations than it is in automated LLM-as-a-judge evaluations", because "maximizing preference likelihood does not mean one is maximizing human utility" [1]. We believe these results provide a key insight for aligning diffusion models with human feedback and are valuable for the community.
>
> If you have any further questions, feel free to let us know in the discussion period.
>
> [1] Ethayarajh, Kawin, et al. "KTO: Model alignment as prospect theoretic optimization." (2024).

---

> > ### Comment · Area_Chair_L2mV · 2024-08-08
> >
> > Could you please specify the number of test prompts (and hence the margin of error) used for Pick-a-Pick, HPS, and Parti prompts?
> >
> > Consider that if A outperforms B 52% of the time with a 95% confidence interval of (50.5%, 53.5%), A is statistically better than B. However, is such a small improvement practically significant? This is the type of question the paper needs to address more clearly.

---

> ### Author Response · Authors · 2024-08-08
> **Further Discussion of "Practical Significance"**
>
> The number of test prompts is **500 for Pick-a-Pick, 3210 in HPS and 1632 in PartiPrompts**. We generate 5 images per prompt. For human evaluation, we collected 300 responses on randomly sampled HPS prompts.
>
> We agree that the "practical significance" of the improvement is very important, which is why we performed the human evaluation. Since human evaluations show considerable improvement over baselines, we argue that our improvement is "practically significant," because the automatic metrics are mere surrogates for human preference (many of these metrics such as PickScore and ImageReward use reward models trained on human preference data).
>
> To further investigate the significance of our improvements, we compare the automatic win-rate of our method against SD v1.5 versus the automatic win rate of Diffusion-DPO against SD v1.5 on HPS, the largest of three datasets. The results are shown in table below
>
> |                         | Aesthetic | PickScore | ImageReward | CLIP | HPS    |
> | ----------------------- | --------- | --------- | ----------- | ---- | --- |
> | Ours vs SDv1.5          | **70.0**     | **74.5**      |      **72.0**       |   53.7   |   **54.2** |
> | Diffusion-DPO vs SDv1.5 | 60.3     | 68.1      |   59.0          |   **54.2**   | 54.1    |
>
>
> It has been well established the improvements of Diffusion-DPO are of "practical significance". Compared with Diffusion-DPO, Diffusion-KTO achieves "significantly" higher win-rates against baseline on Aesthetic (+9.7%), PickScore (+6.4%), Image Reward (+13.0%), and is comparable on HPS (+0.1%) and CLIP (-0.5%).  Given these results, we argue that we are our method also achieves an improvement of "practical significance".
>
> In summary, we believe that "practical significance" is best measured by human evaluation, which show decisive advantages of Diffusion-KTO. Additionally, we also note that when compared with baseline methods, Diffusion-KTO achieves a larger gap than Diffusion-DPO, which is known to have significant improvements.
>
> To avoid further confusions, we would like to clarify that all occurrences of the term “significance” in the above discussion refers to “practical significance” mentioned in previous comments from AC.
>
> If you have additional questions or would like to see additional experiment results, don't hesitate letting us know.

---

> > ### Comment · Area_Chair_L2mV · 2024-08-09
> >
> > Note that I don't have doubt that Diffusion-KTO outperforms SD1.5 with practical significance. My concern on "practical significance" was mainly about:
> >
> > 1. Diffusion-KTO vs SFT
> >
> > 2. Diffusion-KTO vs Diffusion-DPO

---

> ### Comment · Reviewer_YybC · 2024-08-09
> **novelty**
>
> i think even the Diffusion-DPO work can then be viewed in a similar vein, it just used the method from NLP for the text-to-image model. And to a certain extent, this is a valid concern. I suppose the the KTO method is more practical as it does not require paired data but rather text-image pairs that are deemed good to bad which is more realistic.
>
> There is no redeeming in terms of memory / compute usage either.

---

> > ### Comment · Area_Chair_L2mV · 2024-08-09
> >
> > I completely agree that methods not requiring paired data are more practical, and both Supervised Fine-Tuning (SFT) and KTO are appropriate for such scenarios. The key issue is whether KTO-based methods can surpass the performance of SFT. If diffusion-KTO fails to clearly outperform SFT—or only achieves marginal improvements without practical significance—I would question its utility.

---

> ### Author Response · Authors · 2024-08-09
> **practical significance vs SFT baseline**
>
> **Additional Human Evaluation Results**
>
> First, we would like to highlight that **human evaluation shows Diffusion-KTO has a 70.2% win rate against Diffusion-DPO**. We also conducted **additional human evaluation of Diffusion-KTO vs SFT, which results in a win rate of 72.3% in favor of Diffusion-KTO.**
>
>
> **Further discussion on Automatic Metric**
>
> In additional to head-to-head comparison provided in Appendix Table 3, we additionally incorporate the SFT baseline in the results (Method X vs SDv1.5) from previous comment on the HPSv2 test set.
>
> |                         | Aesthetic | PickScore | ImageReward | CLIP | HPS    |
> | ----------------------- | --------- | --------- | ----------- | ---- | --- |
> | Ours vs SDv1.5          | **70.0**     | **74.5**      |      **72.0**       |   53.7   |   **54.2** |
> | Diffusion-DPO vs SDv1.5 | 60.3(-9.7)     | 68.1(-6.4)      |   59.0(-13.0)          |   **54.2**(+0.5)   | 54.1(-0.1)    |
> |SFT vs SDv1.5 |  66.1(-3.9) | 64.7(-9.8)  |  63.4(-8.6) | 49.8(-3.9) | 50.2(-4.0) |
>
>
> Compared with SDv1.5, **Diffusion-KTO achieves a significantly higher win rate than the SFT and Diffusion-DPO baselines.** For example, Diffusion-KTO has a lead of +9.7 on Aesthetic, +6.4 on PickScore, and +13.0 on ImageReward when compared with DPO, and a lead of +3.9 on Aesthetic, +9.8 on PickScore, and +8.6 on ImageReward compared with SFT.
>
> In addition, we would argue **that results presented in Appendix Table 3 and the table above are indeed "practically significant"**. For example, while it may seem that a win rate of 54.0 on ImageReward (Ours vs SFT, appendix Table 3) in a head-to-head comparison is not "significant", comparing (Ours vs SDv1.5) and (SFT vs SDv1.5), we can see that Diffusion-KTO expands the margin of improvement by +8.6. **Comparing with the neutral baseline (0.5 win rate), we expand the gain from (+13.4) of "SFT vs SDv1.5" to a gain of (+22.0) of "Ours vs SDv1.5", or 164% when evaluated by ImageReward.** The gains demonstrated by Diffusion-KTO indicate undoubtedly a significant improvement.
>
> In general, the community has not yet established a clear numerical threshold for what constitutes an automatic win rate of 'practical significance' in text-to-image diffusion models, as these are relatively new metrics. This is why it is important to consider the results provided by human evaluations. In other fields, such as image classification and object detection, **even a 1–2-point gain in Top1-Acc or mAP is considered a meaningful improvement**. If the AC is suggesting that we adopt a specific numerical threshold to determine the 'practical significance' of the improvements presented in Appendix Table 3 and Table 4, we would appreciate clarification on what that threshold should be to facilitate further discussions. Otherwise, we maintain that: **1) human evaluation has demonstrated that our improvement is of 'practical significance,' and 2) Diffusion-KTO provides a more significant margin of improvement (Table above) when compared to SFT or Diffusion-DPO**.
>
> Note: we discovered an editorial oversight in the original manuscript: the columns of the tables are mislabeled. The column label "ImageReward" and "HPS v2" should be swapped for all win rate tables. This has been corrected in the tables we provide above.

---

> > ### Comment · Reviewer_MiLc · 2024-08-09
> > **Clarification**
> >
> > I've read a bit through this thread, and I think the conversation has largely been hindered by miscommunication.
> >
> > I believe the AC is asking the authors if they have a **direct comparison** between Diffusion-KTO model and an SFT model. In particular, this would be a direct win rate of Diffusion-KTO over SFT, rather than a comparison of win rates of these methods versus a base SD 1.5 model (as was just mentioned again). If this is not what the AC is looking for, please correct me.
> >
> > Can the authors provide this? I believe the images are generated from the same prompts, so it would not be too hard to get both the Diffusion-KTO samples and the SFT samples and then compare them side by side.
> >
> > Pros of this evaluation strategy: this avoids an issue that results in the comparisons being non transitive. In particular, it is possible that Diffusion-KTO improves **more** images to a lesser degree than SFT.
> >
> > Cons of this evaluation strategy: this type of pairwise comparison should not be the de facto method for comparing preference optimization strategies for diffusion models for academic papers. While some LLM systems like lmsys do an elo rating, such a system would be impractical for diffusion models (since they are more quite a bit more expensive to sample from). This is **fine** for Diffusion-KTO and SFT since the main benefit of Diffusion-KTO is getting rid of preference pairs, which is semantically similar to SFT and is the key benefit of the method.

---

> > > ### Comment · Area_Chair_L2mV · 2024-08-09
> > >
> > > Thank you, Reviewer MiLc, for your discussion! Yes, you are 100% correct that I was "asking the authors if they have a direct comparison between Diffusion-KTO model and an SFT model."
> > >
> > > The additional information the authors provided was helpful, but was not directly addressing that question.

---

> ### Author Response · Authors · 2024-08-10
> **Direct Comparison**
>
> To reviewer MiLc and AC,
> We agree that there has been some miscommunication and appreciate your efforts to clarify your questions and comments. To clarify, **we have provided direct comparison with SFT baselines.**
>
> 1. We have provided automatic win-rates (Diffusion-KTO vs SFT, Diffusion-KTO vs Diffusion-DPO, etc) in Table 1,  Appendix Tables 3 and 4.
>
> 2. In the comments above, we provided an additional human evaluation which is a direct comparison between images generated by Diffusion-KTO and the SFT baseline using the same HPSv2 prompts. In this human evaluation against SFT, Diffusion-KTO had a win-rate of 72.3%.
>
> 3. Further, we also provided qualitative results in Figure 5, which is a "side by side" comparison of images generated using the same prompts from various baselines.
>
> To our best understanding, we think the discussion is not about whether certain results are missing. Rather, its about how to interpret the results.
>
> We thanks reviewer MiLc's insight and hope the above information is helpful. If you belive additional forms of "direct comparisons" other than (1) (2) (3) can facilitate the discussion, or there are better metrics to measure the "degree of improvement", we are happy to provide those as well.
>
> Conceptually, we would also point out that a key difference between Diffusion-KTO and SFT is that Diffusion-KTO learns to "avoid" bad generations. While this property may be hard to measure directly, in Figure 6 we show that in the 2D Gaussian example, Diffusion-KTO and SFT are both able to fit the distribution of "desirable data", but Diffusion-KTO uniquely avoids "undesirable data".

---

> > ### Comment · Area_Chair_L2mV · 2024-08-10
> >
> > Great, now we are on the same page. Could you respond to my previous questions:
> >
> > - Consider that if A outperforms B 52% of the time with a 95% confidence interval of (50.5%, 53.5%), A is statistically better than B. However, is such a small improvement practically significant? This is the type of question the paper needs to address more clearly.
> >
> > - Note that I don't have doubt that Diffusion-KTO outperforms SD1.5 with practical significance. My concern on "practical significance" was mainly about:
> >
> >    - Diffusion-KTO vs SFT
> >
> >    -  Diffusion-KTO vs Diffusion-DPO
> >
> > More specifically, could you illustrate what the comparisons "Diffusion-KTO vs SFT" and "Diffusion-KTO vs Diffusion-DPO" would look like if you included error bars or a 95% confidence interval in Tables 3 and 4?
> >
> > Additionally, I have two more questions:
> >
> > 1. Regarding the visual results in Figure 5, could you detail the procedure used to select which random generations to display? Given that the winning rates (from both automatic and human evaluations) suggest the superiority of Diffusion-KTO is not overwhelming, the likelihood of it outperforming all others in a single random generation seems low.
> >
> > 2. Figure 6 is quite informative; however, it raises a question about whether you have adequately utilized negative-labeled examples in SFT to steer clear of poor-performing regions. Further clarification on this point would be helpful.

---

> ### Author Response · Authors · 2024-08-11
> **Answer to AC Question #1 out of 2**
>
> **Q: Is such a small improvement practically significant?**
>
> **Response 1: In a hypothetical of 50.5%-53.5, the answer is yes.** We apologize for the confusion in adressing this question. Without any prior literature or community consensus to establish a quantative threshold of "practical significance", we argue that all results that are statistically significant are of "practical use", as it implies "an average improvement". In fact, it is precisely because of the large amount of test data, which leads to low margin of errors, that the ML community consider even 1 or 2 point gain in benchmarks such as ImageNet-1K classification as "significant". We believe, this discussion is outside the scope of our paper and is a broader problem within the ML community in general. To highlight the aspect of "average improvements", we additionally provide average scores on HPS v2 below.
>
> |                         | Aesthetic | PickScore | ImageReward | CLIP | HPS    |
> | ----------------------- | --------- | --------- | ----------- | ---- | --- |
> | SDv1.5          |5.41    | 20.70      |     0.135       |   32.67 |   0.348 |
> | SFT | 5.60     | 21.09      |  0.592          |   32.68  | 0.352   |
> |Ours | **5.66**| **21.36**  | **0.749** | **33.07** | **0.355** |
>
>
>
> **Response 2: We strongly disagree with the AC's premise of "small" improvements**
>
> We would like to respectfully mention, that we have previously inquired about a clear numerical threshold for "practical significance," but this has not been specified. Without a defined standard, it is unfair to justify dismissing our improvements as insignificant. We would like to point out that **we have also presented human evaluation results comparing Diffusion-KTO and SFT side-by-side and found that human evaluators prefer Diffusion-KTO generations 72.3% of the time.** We believe this 70%+ win-rate is practically significant considering human evaluation is the best measurement of how well models are aligned with human preferences. We kindly request the AC to provide a clear threshold and give justifications for such threshold, otherwise, we stand that our results are significant particulary, practically significant.
>
>
> We would like to also point out that the win rate is more thoroughly studied in the context of LLM. At the moment, 54% win rate is the win rate between LLama3.1-405B-instruct versus LLama3.1-70B-instruct on lmsys tournament. We recognize that the task is different, however given the lack of related studies in the context of text-to-image generation, we think this information may be useful.
>
>
> **Response 3: Discussions on specific "in-significant" numbers**.
>
> Looking back at the discussion, the AC noted his/her reservations based on some low numbers in Table 3 and Table 4, but we think our strong results against SFT in Table 1 have been largely marginalized. We apologize but as we mentioned above, our stance is that it is unreasonable to expect Diffusion-KTO to exhibit a large numeric gap on all metrics, as such are rare cases for most works. We believe that all results should be looked at holistically, based on the following principles.
>
> 1. Human evaluation should outweigh automatic ones, as this is the direct objective of preference optimization and human-utility optimization.
> 2. Among all numerical metrics, PickScore is the most significant one as it is trained to reflect the human preference exhibited in the Pick-a-Pick dataset, and is most suitable to evaluate the performance of training algorithms. While other metrics are also trained on preference data, differences in dataset curation may lead to different biases in these reward models.
>
> In this regard, we argue that human evaluation and the results on Pick-a-Pick test set in Table 1 are the best measurements to compare the effectiveness of optimization algorithms, which the AC has kindly acknowledged: "clearly higher winning rates based on Pick-a-Pick test prompts".

---

> ### Author Response · Authors · 2024-08-11
> **Answer to AC Question #2 out of 2**
>
> As requested, to facilitate the discussion, we listed the error bars of 95% confidence interval as below. We observe that all win rate has a margin of error <2%.
>
>
> Win rate on Pick-a-Pick Test Set (2500 generations)
>
> |                       | Aesthetic        | PickScore          | ImageReward | CLIP | HPS    |
> | ----------------------- | --------- | --------- | ----------- | ---- | --- |
> | Ours vs SFT         |56.0$\pm 1.95$  | 73.0$\pm 1.74$   |     60.6 $\pm 1.92$        |   57.6 $\pm 1.94$      |   57.2 $\pm 1.94$     |
> | Ours vs Diffusion-DPO | 70.4 $\pm 1.79$    | 59.6   $\pm 1.92$  |   74.2$\pm 1.72$  | 48.4$\pm 1.96$ | 49.2 $\pm 1.96$ |
>
> Win rate on HPS v2 Test Set (16050 generations)
>
> |                         | Aesthetic | PickScore | ImageReward | CLIP | HPS    |
> | ----------------------- | --------- | --------- | ----------- | ---- | --- |
> | Ours vs SFT         | 54.1 $\pm 0.77$  | 61.2$\pm 0.76$   |      54.0 $\pm 0.77$        |   54.6 $\pm 0.77$      |   54.2 $\pm 0.77$     |
> | Ours vs Diffusion-DPO | 58.8 $\pm 0.76$    | 58.6   $\pm 0.76$  |   63.9 $\pm 0.74$          |   48.7 $\pm0.77$   | 48.8  $\pm0.77$   |
>
>
> Win rate on PartiPrompts (8160 generations)
>
> |                         | Aesthetic | PickScore | ImageReward | CLIP | HPS    |
> | ----------------------- | --------- | --------- | ----------- | ---- | --- |
> | Ours vs SFT         | 50.3 $\pm 1.1$  | 58.6 $\pm 1.08$  |     54.7 $\pm  1.09$        | 54.2 $\pm 1.09$      |   51.4 $\pm  1.10$     |
> | Ours vs Diffusion-DPO | 59.5 $\pm 1.08$    | 48.3 $\pm 1.10$  |   59.3$\pm  1.08$          |   48.6 $\pm  1.10$   | 48.2  $\pm  1.09$  |
>
> **Q:How do you sample prompts for Fig 5**.
> To highlight the advantage of Diffusion-KTO in real-world scenarios, we create prompts based on user prompts shared over the internet. In particular, we heavily referenced Playground.AI, where users share generated images alongside their prompts. While these prompts are manually written, we believe they reflect typical use cases in real-world scenarios and is generally similar to the "good" samples in the Pick-a-Pic dataset, which is also written by human labelers. To further highlight the fairness of our method, we report the automatic win rate of 20 randomly selected prompts and generate 5 images per prompt. The prompts are copied from the front page of playground AI on Aug 10, 2024. An example prompt would be
>
> ```
> Photorealistic wet-erase marker portrait depicts Cleopatra with a hybrid aesthetic of a bikini and Galadriel's queenly regalia,....
> ```
>
> The results are as follows
>
> |            Metric       |  |
> | ----------------------- | --------- |
> | P(Ours > SDv1.5)          |  0.75 |
> | P(Ours > SFT)          | 0.75 |
> | P(Ours > DPO)   | 0.70 |
> | P(Ours > SFT \| Ours > DPO)   | 0.785 |
> | P(Ours >  SDv1.5 \| Ours > SFT,Ours > DPO)          | 0.909 |
>
>
> For simplicity, we adopt an overall win rate in the above calculation. X > Y if and only if amongst 5 metrics, (Aesthetic ,PickScore ,ImageReward , CLIP ,HPS), at least 3 are better. **Contrary to AC's statement, "the likelihood of (Diffusion-KTO) outperforming all others", in the distribution of user-specified prompt sampled from Playground AI, is 49.9%, which is not small.** This is because the win rates are highly correlated with each other and not independent. For example, if Diffusion-KTO beats SFT and DPO on a prompt, it is likely that it will beat SDv1.5 as well.
>
>
> **Q: Have you adequately utilized negative-labeled examples in SFT**. We request further clarification on how to adequately utilize negatives in SFT as it is not in the default formulation. We tried to naively add a negative term to the L2 loss to "steer clear of poor-performing regions" using the loss $\mathbb{E}[y(x_t)\lVert \epsilon_\theta(x_t)-\epsilon\rVert^2]$ where $y(x_t)\in\{+1,-1\}$ is the label. The model diverge to infinity as the loss on positive samples has range (0,+inf), but the loss on negative samples has range (-inf,0). This results in the model pushing the loss to -inf.  In fact, it is precisely because this form of naive contrastive SFT does not work, that the community has resorted to alternative approaches such as RL, Inverse RL, and more recently DPO.
>
> In summary, we want to apologize for the confusion during this discussion and we hope in this response to have cleared some of the questions by providing detailed information.

---

> > ### Comment · Area_Chair_L2mV · 2024-08-12
> >
> > As the AC for this paper, I have consistently urged you to substantiate the empirical advantages of your approach, especially in light of the novelty concerns of the reviewers I highlighted based on initial reviews. As previously stated, "Given these novelty concerns, the empirical performance improvements presented will be crucial in informing my decision. I have reservations about the significance of these performance gains and would appreciate further insights from both reviewers and authors on this matter."
> >
> > I appreciate the authors providing additional results and emphasizing that their human evaluations clearly prefer Diffusion-KTO over the baselines. I am glad to accept that advantages, but I'd like to also point out that while human evaluation is invaluable, it also poses challenges in terms of reproducibility and comparative cost.
> >
> > My primary concerns were initially centered around Tables 3 and 4, which prompted my inquiries about the "statistical significance" of the improvements and whether the differences were substantial enough to have practical significance. I am pleased that the authors are now providing error bars for Tables 3 and 4, indicating that some improvements are statistically significant, although it remains challenging to draw meaningful conclusions for some metrics.
> >
> > The authors suggest focusing on Table 1, noting that "PickScore is the most significant metric as it is trained to reflect human preferences exhibited in the Pick-a-Pick dataset and is most suitable for evaluating the performance of training algorithms. Other metrics, trained on preference data, may also be relevant, but differences in dataset curation could introduce biases in these reward models." However, it's important to consider that the method's advantages might be less pronounced under domain shifts (shifting from Pick-a-Pick to other prompts).
> >
> > Regarding Figure 5, the selection criteria for these results remain unclear to me. Were these images chosen by finding a specific seed where Diffusion-KTO outperformed all other models, and then displaying those particular pictures?
> >
> > Regarding the use of negative labels, I find your response satisfactory. Nevertheless, for the toy example illustrated, I think integrating a relatively straightforward technique, such as classifier guidance, into the diffusion model could significantly enhance the model's ability to steer clear of undesired regions.

---

> ### Author Response · Authors · 2024-08-12
>
> Hi,
>
> We are glad that the AC recognize these advantages. We sincerely apologize for any potential misunderstanding and miscommunications. We might have wrongly assumed the discussion around "practical significance" was based on perceived "substantial differences in numbers" as a criterion in lieu of "statistical significance", a stance we found to be not objectively grounded.  We are glad that this is not the case, and the confusions have been cleared away.
>
> **Q:Further Clarification on Fig 5**
>
> For Figure 5, **the images are not chosen by specific seeds, we just randomly generate an image per prompt**. However, the prompt, which is listed on the left most column, are not randomly selected from a dataset. They are manually written ones based on practical use cases on Playground AI, a website where users share their prompts. Typically, these prompts are highly specific.  Our observations show that given these highly specific prompt, (we have provided an example above), the images generated by each model will have small variations. Hence, the difference between models is fairly consistent across generations. We provided additional analysis of these prompts in the previous comment. Please feel free to let us know if you have any additional questions.
>
> **Q: Reproducibility of Human Evaluation**
>
> We agree that this is a limitation for human evaluations. We hope future works from the community can establish an open and reproducible ranking based on human evaluation like lmsys for LLMs. However, at the moment, we have provided a detailed description of the human evaluation protocol in appendix A.2, alongside designs to mitigate systematic biases (e.g. "not given any information about which methods are being compared and the order of methods (left or right) is randomized").  The statistic confidence which we provided in previous comments and our rebuttal to reviewer 7SUT in this case should be a good measurement for reproducibility.
>
> **Q:It's important to consider that the method's advantages might be less pronounced under domain shifts.**
>
> We agree with this observation. Then, it is critical to consider what is the distribution of prompts that best reflect a "practical use", as this is a key focus of this discussion.
>
> Consider the common use case of generating people as a case study. The **PartiPrompts** "People" category contains prompts such as "a woman with long hair", "a boy going to school".
>
> On **HPS v2**, examples of prompts involving people are "Blond-haired girl depicted in anime style", "A cyberpunk woman on a motorbike drives away down a street while wearing sunglasses"
>
> **Pick-a-Pick** contains a mix of different styles and even in different languages. Examples are "A woman’s wrist with a thin golden bracelet, romantic city lights in the background", "Eighteen year-old girl, pale skin, smooth skin, black jacket, ......  Chris Bachalo, Belén Orteg".
>
> Most prompts on **Playground AI** are similar to longer ones in Pick-a-Pick, such as "Transparent-skinned creature showing complex anatomical structures intertwined with botanical patterns,... Lynn Maycroft's work, cinema."
>
> Unlike the previous three academic datasets, these are prompts used by actual paid subscriber who are willing to pay 15-35 USD per month for T2I models. The authors believe this distribution best illustrate the "practical usecase", which is why we use this for our qualitative results in Fig5. While we also made our best attempt to provide some quantitative analysis in previous comments, various engineering and legal complications of large-scale web-scrapping prevents us from scaling these analyses to a full-sized benchmark.
>
> In summary, we agree that "advantages might be less pronounced under domain shifts", but we consider this a natural consequence of training on Pick-a-Pick. Given that KTO only requires binary preferences, it is easier to collect such preference data on the intended distribution (e.g. PartiPrompts). The authors hold that results presented in this paper have met the burden of "substantiate the empirical advantages". It can be reasonably expected that KTO should show similar improvements with proper preference data on any distribution of prompts.
>
> The main contribution of Diffusion-KTO is a general and effective optimization algorithm, not the actual model trained on Pick-a-Pick dataset. The latter is an experiment to study the effectiveness of the former. We do not claim this specific model weight has stronger generalization capability on unseen concepts, but rather, it can reasonably generalize to "practical usecases" as discussed above. In fact, we believe it is unrealistic to expect any model (including SFT,DPO) that is trained on Pick-a-Pick prompts to generalize to prompts such as  "element", "concurrent lines" (which are concepts uniquely present in PartiPrompts). It is even debatable if human preferences make sense for such prompts. We appreciate the AC's detailed feedback and discussion.

---

> > ### Comment · Area_Chair_L2mV · 2024-08-13
> >
> > I appreciate the authors' clarification on these subtle yet important points. I strongly encourage you to incorporate these discussions into your camera-ready version, as I believe this will help alleviate any doubts and enhance understanding.

---

> > > ### Author Response · Authors · 2024-08-13
> > >
> > > Hi
> > > We thank the AC's constructive feedbacks and will provide these important details (e.g. statistic significance, details of Fig 5) in the camera-ready version. We find these discussions exceptionally helpful and will revise our paper accordingly.

---

### Decision · Program_Chairs · 2024-09-25

**Decision:**

Accept (poster)

**Comment:**

The paper introduces Diffusion-KTO, a method largely seen as a straightforward extension of KTO, which was originally applied to fine-tuning language models, now adapted for fine-tuning diffusion models. This approach could also be considered a variant of Diffusion-DPO, with the DPO component replaced by KTO. From a methodological standpoint, the novelty is somewhat limited. However, the paper is notable for being among the few that explore human preference-based optimization of diffusion models. It provides effective results and detailed comparisons with established methods, earning the favor of all four reviewers for acceptance.

During the post-rebuttal discussion phase, there was an in-depth dialogue between the Area Chair and the authors, focusing on areas for improvement. Key issues raised during the discussion included questioning the statistical significance and practical utility of the minor improvements reported in some of the tables, as well as acknowledging potential performance degradation due to domain shifts in the text prompts. The authors are encouraged to carefully consider these points when preparing the camera-ready version of the paper.